



# Particle shapes and infrared extinction spectra of nitric acid dihydrate crystals: Optical constants of the β-NAD modification

Robert Wagner[1], Alexander D. James[2], Victoria L. Frankland[3], Ottmar Möhler[1], Benjamin J. Murray[4], John M. C. Plane[2], Harald Saathoff[1], Ralf Weigel[5], and Martin Schnaiter[1]

[1]Institute of Meteorology and Climate Research, Karlsruhe Institute of Technology, Karlsruhe, Germany
[2]School of Chemistry, University of Leeds, Leeds, UK
[3]School of Chemistry and Chemical Engineering, University of Surrey, Guildford, UK
[4]School of Earth and Environment, University of Leeds, Leeds, UK
[5]Institute for Atmospheric Physics, Johannes Gutenberg Universität Mainz, Mainz, Germany

*Correspondence to*: Robert Wagner (robert.wagner2@kit.edu)

**Abstract.** Satellite- and aircraft-based mid-infrared measurements of polar stratospheric clouds (PSCs) have provided spectroscopic evidence for the presence of β-NAT (nitric acid trihydrate) particles. Metastable nitric acid hydrate phases such as α-NAT and α-NAD (nitric acid dihydrate) have been frequently observed in laboratory experiments, but not yet detected

as a constituent of PSCs in atmospheric measurements. As for the β-NAD modification, its formation was first observed in X-ray diffraction measurements when the low-temperature α-NAD phase was warmed to a temperature above 210 K. Its infrared spectrum has been reported, but so far no optical constants have been derived that could be used as input for infrared retrievals of PSC composition. In this work, we show that β-NAD particles were efficiently formed in isothermal, heterogeneous crystallisation experiments at 190 K from supercooled $HNO_3/H_2O$ solution droplets containing an embedded mineral dust or

meteoric smoke particle analogue. An inversion algorithm based on a T-matrix optical model was used to derive for the first time the mid-infrared complex refractive indices of the β-NAD modification from the measured extinction spectrum of the particles. In contrast to the heterogeneous crystallisation experiments, the α-NAD phase was formed when the $HNO_3/H_2O$ solution droplets did not contain a solid nucleus and crystallised homogeneously. Using a light scattering detector that recorded two-dimensional scattering patterns of the crystallised NAD particles, we were able to determine predominant shapes of the

α- and β-NAD crystals. We found that α-NAD grew into elongated, needle-shaped crystals, while β-NAD particles were compact in shape. This agrees with previously reported images of α- and β-NAD particles grown on the cryo-stage of an Environmental Scanning Electron Microscope.

## 1 Introduction

Laboratory studies, field campaigns, and space-borne observations have led to a comprehensive knowledge of the formation,

occurrence, and composition of polar stratospheric clouds (PSCs) (Lowe and MacKenzie, 2008; Peter and Grooß, 2012;





Molleker et al., 2014; Tritscher et al., 2021). An important constituent of PSCs are crystalline nitric acid hydrates, in particular nitric acid trihydrate (NAT) and nitric acid dihydrate (NAD), both of which exist in two different phases, called the α- and β-modifications (Koehler et al., 1992; Toon et al., 1994; Lebrun et al., 2001a; Lebrun et al., 2001b; Tizek et al., 2002; Grothe et al., 2008; Iannarelli and Rossi, 2015). For each compound, the α-phase denotes the low-temperature modification that

irreversibly transforms into the more stable β-phase upon annealing at higher temperature. The trihydrate NAT, along with the monohydrate NAM, is one of the two thermodynamically stable hydrate phases of the $HNO_3/H_2O$ system (Lebrun et al., 2001b; Beyer and Hansen, 2002), and the first detection of NAT in the polar vortex was carried out with a balloon-borne mass spectrometer (Voigt et al., 2000). Mid-infrared limb-emission measurements with the satellite-based spectrometer MIPAS (Michelson Interferometer for Passive Atmospheric Sounding) identified which of the two NAT modifications was

predominant. The distinctive PSC spectral signature at a wavenumber of 820 $cm^{-1}$, corresponding to the $\nu_2$ symmetric deformation mode of the $NO_3^-$ ion (Spang and Remedios, 2003), could be best matched by refractive index data for β-NAT (Höpfner et al., 2006). The β-NAT phase was also identified in broadband infrared transmittance spectra recorded between 4400 and 750 $cm^{-1}$ with the Fourier transform spectrometer on board the Atmospheric Chemistry Experiment (ACE) satellite (Lecours et al., 2022; Lecours et al., 2023).


Homogeneous nucleation of NAT particles from supercooled ternary solution droplets (STS; $H_2SO_4/HNO_3/H_2O$) has been considered to proceed too slowly under stratospheric conditions to explain the observed NAT number densities in the stratosphere (Koop et al., 1995; Knopf, 2006). Instead, the heterogeneous nucleation of NAT on pre-existing ice particles is a widely accepted formation mechanism (Tritscher et al., 2021; Weimer et al., 2021), supported e.g. by the detection of NAT

particles downwind from mountain wave induced ice clouds (Carslaw et al., 1998). Furthermore, there is growing evidence for an additional ice-free NAT formation pathway (Pagan et al., 2004; Voigt et al., 2005), which could involve the heterogeneous nucleation of NAT on refractory aerosol particles (Bogdan and Kulmala, 1999; Hoyle et al., 2013; James et al., 2018; James et al., 2022). The origin of these particles includes the input of meteoric ablation material from the upper stratosphere and mesosphere (Murphy et al., 2014; Weigel et al., 2014; Ebert et al., 2016; Schneider et al., 2021).


Although only β-NAT - but no metastable phase such as α-NAD, β-NAD, or α-NAT - has been observed in the stratosphere so far, it is not certain whether β-NAT must necessarily be the polymorph that crystallises first by either homogeneous or heterogeneous nucleation. This is a consequence of Ostwald's step rule, which states that the formation of the thermodynamically stable polymorph can proceed via various intermediate, metastable states, whose crystallisation may have

a lower kinetic barrier due to a better structural match with the short-range order of the supersaturated liquid phase (Ostwald, 1897). The kinetic stability of a certain metastable phase in the $HNO_3/H_2O$ system depends on the temperature, the stoichiometry of the supersaturated liquid phase, and other factors such as stabilising effects due to an ice surface (Tizek et al., 2002; Tizek et al., 2004; Grothe et al., 2008; Weiss et al., 2016). In a study using an aerosol flow tube, supercooled $HNO_3/H_2O$



solution droplets of 1 : 3 molar stoichiometry were first cooled to 153 K and then warmed to 178 K to accelerate the crystal
growth of the nuclei formed homogeneously at 153 K (Bertram and Sloan, 1998b). This technique allowed the infrared
extinction spectrum of the completely crystalline particle ensemble to be recorded at 178 K, showing a good match with a Mie
theory calculation using the optical constants for α-NAT (Richwine et al., 1995). Recent work emphasised the importance of
the α-NAT phase also with respect to the heterogeneous nucleation pathway involving ice particles (Weiss et al., 2016). This
study showed that α-NAT could be easily grown on a water ice film at temperatures below 188 K due to the absence of a
significant nucleation barrier. Upon warming, α-NAT completely converted to β-NAT within a narrow temperature range of
187.5 to 195 K (Weiss et al., 2016). A considerable number of aerosol chamber and flow tube studies have also demonstrated
the nucleation of NAD particles (Barton et al., 1993; Disselkamp et al., 1996; Bertram and Sloan, 1998a; Prenni et al., 1998;
Tisdale et al., 1999; Möhler et al., 2006a; Stetzer et al., 2006; Iannarelli and Rossi, 2015). The occurrence of this nucleation
process is understandable due to the lower nucleation barrier for NAD formation compared to the high saturation ratios required
for NAT nucleation (Salcedo et al., 2001). In the vast majority of cases, these studies found the formation of the α-NAD
modification, which can be explained by the fact that the α-NAD structure corresponds better to the short-range order of the
supersaturated liquid phase than the three-dimensional network of H-bonds characteristic of β-NAD (Grothe et al., 2004). Only
the infrared spectra presented in Tisdale et al. (1999) of NAD particles formed below 160 K, and annealed at temperatures
between 160 and 221 K, clearly showed the spectral signatures later ascribed to β-NAD (Grothe et al., 2004).

In addition to the question of which polymorphic form predominates, the particle shape of the nitric acid hydrates is another
important factor as it can influence the vertical redistribution of HNO₃ in the winter stratosphere by changing the settling
velocities of the particles (Grooß et al., 2014; Woiwode et al., 2014). The peak-like signature of the $\nu_2$ nitrate mode at 820
cm$^{-1}$ analysed by Höpfner et al. (2006) was best fit with compact particle shapes, representing high densities of small β-NAT
particles with radii < 3 μm. Further aircraft- and space-borne MIPAS measurements revealed a modified, shoulder-like
signature at 820 cm$^{-1}$ first observed in local Arctic PSC flights and later detected as a vortex-wide signature in the 2011/12
Arctic winter stratosphere (Woiwode et al., 2016; Woiwode et al., 2019). This signature could only be accurately fitted for
larger (equal-volume sphere radii of 3 – 7 μm) and highly aspherical β-NAT particles, which have needle- or platelet-like
shapes with aspect ratios (AR) of 1 : 10 and 10 : 1, respectively. The best overall agreement between simulation and
measurement was achieved for elongated, needle-shaped particles with AR = 0.1, which is supported by laboratory experiments
showing that needles can develop when β-NAT grows at an ice boundary at 193 K (Grothe et al., 2006). Highly aspherical
particle shapes could also explain the unexpectedly large optical diameters (in some cases larger than 20 μm) of HNO₃-
containing particles detected with aircraft-borne optical in situ instruments (forward scattering and optical array imaging
probes) (Molleker et al., 2014). For compact particle shapes, such sizes could not be reconciled with simulations based on
realistic values for the sedimentation velocity and the time available for particle growth (Molleker et al., 2014). However, they
can now likely be interpreted as the maximum dimension of highly elongated β-NAT particles (Molleker et al., 2014; Woiwode





et al., 2016). Tritscher et al. (2021) argued that although particle growth probably starts with a nearly spherical geometry, further growth in slightly supersaturated vapours can lead to a higher degree of asphericity, resulting in different particle shapes depending on the nucleation mechanism and growth history.


The same effect has been observed in our previous AIDA (Aerosol Interaction and Dynamics in the Atmosphere) cloud chamber study on the homogeneous nucleation of α-NAD particles from aqueous $HNO_3/H_2O$ solution droplets (Wagner et al., 2005a). The infrared spectrum of α-NAD particles generated by shock freezing of a gas mixture of nitric acid and water could be accurately fitted using Mie theory with tabulated optical constants of α-NAD (Toon et al., 1994; Niedziela et al., 1998),

pointing to compact, near-spherical particle shapes. In contrast, homogeneous nucleation of α-NAD from populations of slowly evaporating supercooled $HNO_3/H_2O$ solution droplets at 193 and 195 K over a time scale of 4 h yielded markedly different α-NAD infrared extinction spectra, which could no longer be reconciled with Mie theory. In particular, we observed a strong change in the relative peak intensities of the characteristic nitrate doublet feature with components at about 1450 and 1280 cm$^{-1}$. These spectral changes could be modelled satisfactorily when strongly aspherical particle shapes were considered

in the calculation of the α-NAD extinction spectrum, i.e., AR ≤ 0.2 for prolate and ≥ 5 for oblate shapes (Wagner et al., 2005a). We concluded that the platelet-like, oblate shapes matched the measured extinction spectra slightly better than the needle-like, prolate ones, but did not have an instrument to directly measure particle shape at that time. A few years later, the growth of needle-like α-NAD crystallites was observed on the cryo-stage of an ESEM (Environmental Scanning Electron Microscope) (Grothe et al., 2008). With regard to the characterisation of airborne α-NAD particles, which grew to median equal-volume

sphere diameters of about 2 – 3 μm in our previous studies, we highlighted the possibility of growing them in future experiments to sizes larger than 10 μm, so that their shape could be resolved by a commercial cloud particle imager (Wagner et al., 2005a). However, further instrumental developments such as the Small Ice Detector 3 (SID-3), which measures high-resolution two-dimensional scattering patterns, make it possible to circumvent the size limitation of imaging probes and obtain shape information even for particle sizes below 10 μm (Ulanowski et al., 2014; Vochezer et al., 2016). In the first part of this

article, we report the repetition of the homogeneous nucleation experiment from our earlier study, but now including particle characterisation with the SID-3, in order to finally answer the question of the predominant shape of homogeneously nucleating α-NAD particles that grow slowly in a supersaturated environment.

In the second and central part of this article, we look at a different type of experiment to investigate one of the heterogeneous

nucleation mechanisms of nitric acid hydrates. Specifically, we targeted the crystallisation of $HNO_3/H_2O$ solution droplets with an embedded refractory component that was either a mineral dust particle (illite SE, Arginotec) or a surrogate for meteoric ablation material (sol-gel synthesised olivine, $MgFeSiO_4$). Illite SE is a mixture of several minerals, dominated by illite (Pinti et al., 2012), hereafter referred to simply as "illite". Nucleation activities of meteoric materials for the formation of nitric acid hydrates have already been measured with a drop freezing assay (James et al., 2018; James et al., 2022). While the onset of



melting of the frozen droplets was mostly in agreement with the NAT/ice eutectic, some experiments indicated the melting or recrystallisation of metastable phases (James et al., 2022). The nature of these metastable phases, or that of the phase that first nucleated from the solution droplets, was not the focus of these studies. In the AIDA chamber, the infrared extinction and SID-3 measurements allow an unambiguous identification of the phase and shape of the nucleated nitric acid hydrate particles. We will show that the heterogeneous nucleation experiments led to crystallisation and growth of compact β-NAD particles.

To our knowledge, direct nucleation of β-NAD has not yet been observed in experiments with suspended particles. Furthermore, β-NAD is the only polymorph whose infrared optical constants have not yet been determined. We therefore present an analysis to derive the refractive index data set for β-NAD from our measured infrared extinction spectrum, following an approach recently applied to crystalline ammonium nitrate particles (Wagner et al., 2021). Although not yet identified in the polar stratosphere, our experiments further support the assumption that the formation of metastable NAD phases should

not be excluded a priori. The new optical constants should thus fill a gap for the analysis of current and future infrared measurements of polar stratospheric clouds.

Our article is organised as follows: Section 2 summarises our experimental and theoretical methods, including the instrumentation of the AIDA chamber and the general procedure of the crystallisation studies (Sect. 2.1), the image analysis

of the SID-3 scattering patterns (Sect. 2.2), and the retrieval scheme for determining the infrared optical constants of β-NAD (Sect. 2.3). In the results section, we first give a general overview of the AIDA data for the two types of experiments that yielded different NAD polymorphs, that is α-NAD in the homogenous and β-NAD in the heterogeneous nucleation experiments (Sect. 3.1). Section 3.2 describes the predominant shape of the α-NAD particles based on the analysis of the SID-3 measurements, while Sect. 3.3 then presents the new data set of optical constants for β-NAD. A summary with an

outlook on future chamber experiments concludes our article (Sect. 4).

## 2 Methods

### 2.1 Instrumentation and experimental procedure

The AIDA aerosol and cloud chamber is a coolable and evacuable aluminium chamber with a volume of 84 m³ (Fig. 1). Its scope of application ranges from dynamic expansion cooling experiments on cloud formation on time scales of a few minutes,

to long-term studies on aerosol particle nucleation and ageing on time scales of several days, both at ambient temperatures and conditions of the upper troposphere and lower stratosphere, with a minimum achievable temperature of about 180 K (Wagner et al., 2006b). The experiments presented here are largely based on the procedure and instrumentation used in our earlier studies on the homogeneous nucleation of α-NAD from binary supercooled $HNO_3/H_2O$ solution droplets (Wagner et al., 2005a; Möhler et al., 2006a; Stetzer et al., 2006). In the following, we will briefly summarise this setup and include the novel aspects

of our current work, i.e., the addition of refractory particles to induce heterogeneous nucleation.



As a first step, the insulating containment that houses the AIDA chamber was cooled within 24 h from room temperature to about 190 K, where most of the experiments described in this article were performed. The gas temperature was calculated as the average value from four thermocouple sensors mounted at different heights inside the chamber, with an estimated uncertainty of ± 0.3 K. Before each experiment, the chamber was cleaned by evacuating it to a pressure < 1 hPa, flushing it several times with particle-free and dry synthetic air, and refilling it with the synthetic air to ambient pressure. The initial relative humidity of the chamber air with respect to ice ($RH_{ice}$) was about 10 %, given as the percentage of the ratio between the water vapour pressure measured directly with an in situ tuneable diode laser (TDL) absorption spectrometer (Fahey et al., 2014) and the saturation water vapour pressure of ice at the mean AIDA gas temperature (Murphy and Koop, 2005). The relative uncertainty of $RH_{ice}$ was estimated to ± 5 % (Fahey et al., 2014). A mixing fan mounted about 1 m above the floor of the chamber was in operation for the entire duration of the experiments, resulting in a mixing time of 1.5 minutes.

To generate pure $HNO_3/H_2O$ solution droplets for the homogeneous crystallisation experiments that yielded α-NAD, we prepared carrier gases of water and nitric acid by passing two synthetic air flows over flasks filled with ultrapure water (Barnstead NanoPure System, Thermo Scientific) and concentrated nitric acid (65%, Merck). The mixed carrier gases were injected into the AIDA chamber through a heated Teflon tube. The individual flow rates were adjusted to yield a molar ratio of $H_2O$ to $HNO_3$ of about 3 : 1 in the gas mixture. Upon entering the cold chamber interior, the now supersaturated nitric acid – water gas mixture condensed into supercooled $HNO_3/H_2O$ solution droplets by binary homogeneous nucleation. Injection was continued until an aerosol mass concentration of about 2.2 – 2.7 mg/m³ was reached, which provided a sufficient reservoir to ensure a good signal-to-noise ratio in the infrared measurements for the long-term observation of the slowly evaporating solution droplets that eventually led to the crystallisation of α-NAD particles after a couple of hours. For the experiments on heterogeneous nucleation that yielded β-NAD, we admixed a gas flow that contained the dispersed illite or $MgFeSiO_4$ particles to the gaseous nitric acid–water mixture. This ensured that a part of the $HNO_3$ and $H_2O$ gases condensed onto the solid particles as they entered the cooled AIDA chamber, resulting in a certain fraction of internally-mixed solid-liquid particles whose crystallisation behaviour was markedly different from that of the pure liquid $HNO_3/H_2O$ solution droplets. A rotating brush generator (PALAS, RBG 1000) was used to disperse the illite and the amorphous $MgFeSiO_4$ powder sample. The latter was synthesised using a sol-gel technique (Frankland et al., 2015; James et al., 2017). The number concentration of the co-added solid seed aerosol particles was measured with a condensation particle counter (model CPC3010, TSI) connected to a heated sampling line. Aerosol size distribution measurements will be described later in this article.

The temporal evolution of the $HNO_3/H_2O$ droplet mode (with and without seed aerosol particles) and the crystallisation of nitric acid hydrates were observed by using four types of instruments. They either provided in situ measurements or sampled the particles and probed them inside the cooled insulating containment at the same temperature as inside the AIDA chamber. The instrument setup included an infrared spectrometer, a device for light scattering and depolarisation measurements, an





optical particle counter (OPC), and the SID-3 instrument. In the following, we give a brief description of the first three measurement techniques, whereas the operation of the SID-3 and the analysis of its measurement data are described in detail in Sect. 2.2. The use and mutual complementarity of the individual instrumental techniques is further discussed in Sect. 3.1, where we show time series of the AIDA data for the homogeneous and heterogeneous crystallisation experiments.

The infrared extinction measurements were carried out in situ by coupling a Fourier Transform Infrared spectrometer (IFS66v, Bruker) with an internal multiple reflection cell set to an optical path length of 197 m (Wagner et al., 2006a). The spectra were recorded at wavenumbers between 6000 and 800 cm$^{-1}$ with a resolution of 4 cm$^{-1}$. In the initial phase of the experiment, the infrared spectra could be analysed to determine the composition and mass concentration of the supercooled solution droplets (Stetzer et al., 2006), based on Mie calculations using the temperature- and composition-dependent optical constants of

$HNO_3/H_2O$ (Norman et al., 1999). In the later period, the crystallisation of the particles could be clearly detected by changing frequencies and intensities of the vibrational modes during the transition from the liquid to the solid phase (Wagner et al., 2005a). The detailed steps that were necessary to retrieve the optical constants of β-NAD from its measured infrared extinction spectrum are described in Sect. 2.3. The light scattering and depolarisation measurements were equally carried out in situ with the so-called SIMONE instrument, which mimics the atmospheric polarisation lidar technique inside the AIDA chamber

(Schnaiter et al., 2012). In this setup, the linearly polarised light of a continuous-wave semiconductor laser with an emission wavelength of 488 nm was guided horizontally through the aerosol vessel and absorbed by a beam dump on the opposite chamber side. The light backscattered from the aerosol particles in the centre of the chamber was collected by a telescope optics at a scattering angle of 178°. A Glan-laser prism separated the backscattered light into its parallel and perpendicular components with respect to the polarisation state of the incident laser, and the respective scattering intensities were measured

by two independent photomultipliers. Taking into account the relative sensitivities of the two individual photomultipliers, the intensity ratio of the perpendicular and parallel components yielded the near-backscattering linear depolarisation ratio, δ. While δ is zero for light scattering by homogeneous, spherical solution droplets, aspherical particles such as crystalline nitric acid hydrates cause non-zero depolarisation, with the exact magnitude revealing a complex relationship with particle size and aspect ratio (Mishchenko et al., 1996). Most importantly, the magnitude of δ cannot be simply related to the degree of asphericity,

which means that δ is not necessarily higher the more the aspect ratio of the particles deviates from unity. For example, it has been shown that highly aspherical particles have only a weak depolarisation capability (Zakharova and Mishchenko, 2000, 2001), an aspect we will return to when discussing the δ values observed for the α- and β-NAD particles in our homogeneous and heterogeneous nucleation experiments.

Both the OPC sensor and the SID-3 instrument were mounted in the free space between the bottom of the cloud chamber and the floor of the insulating housing (see Fig. 1). Vertical connecting tubes were used to minimise particle transport losses. The OPC sensor (welas, Palas GmbH) measured the intensity and pulse duration of the light scattering signals caused by individual





particles as they passed through its measurement cell. Based on the number of pulses, their duration, and the volume of the measurement cell, the total number concentration of particles detected could be calculated with an estimated uncertainty of ± 20% (Möhler et al., 2006b). Furthermore, based on Mie calculations that took into account the spectrum of the incident light, the scattering geometry of the sensor, and the refractive index of the particles, the intensity of the scattering signal was assigned to an optical particle diameter. For a refractive index of 1.45, the size classification ranged from 0.45 to 24.0 μm. For aspherical particles such as nitric acid hydrates, these diameters can only be interpreted as apparent sizes, as the scattering phase function is a function of particle shape and also depends on the orientation of the particles in the detection volume (Benz et al., 2005).

## 2.2 The SID-3 instrument: Operation and analysis of the scattering images

The Small Ice Particle Detector Mk.3 (SID-3) is an airborne single particle probe developed by the University of Hertfordshire, UK. It is part of the SID family used to discriminate between supercooled liquid droplets and ice particles in mixed-phase clouds (SID-1) (Hirst et al., 2001), as well as to characterise the size, shape, and structural complexity of small ice crystals and large aerosol particles (SID-2 and SID-3) (Cotton et al., 2013; Schnaiter et al., 2016). The basic measurement concept of SID-3 is to acquire two-dimensional (2-D) light scattering patterns of individual particles in the 7° to 23° forward angular range with a high resolution of better than 0.1°. This is realised by capturing a full azimuthal scattering annulus by an intensified charged coupled device camera (ICCD) equipped with a Fourier lens system. Since coherent laser light with a wavelength of 532 nm is used in SID-3, the forward scattering pattern contains a significant contribution from light diffraction that is analysed to classify the particle shape (Fig. 2). Details of the SID-3 image analysis and shape classification software can be found in Vochezer et al. (2016). In this work we used SID-3 to determine the aspect ratio of α- and β-NAD particles that frequently showed two-fold symmetrical scattering patterns with clearly countable diffraction maxima along the two main axes of the particle (Figs. 2a and b). Representative scattering patterns were selected from the individual experiments and manually inspected to deduce the aspect ratio of the particle as the number ratio of diffraction maxima that were counted along the two symmetry axes of the scattering pattern. In this analysis, the SID-3 ICCD images are interpreted as far field diffraction patterns generated from 2-D obstacles, i.e., rectangles or ellipses. In fact, far field diffraction pattern simulations assuming simple 2-D obstacles nicely mimic the measured patterns (Figs. 2c and d). However, a discrimination between 3-D particle shape attributes like "prolate" and "oblate" is not possible based on this analysis.

The SID-3 instrument was installed into a vacuum-sealed stainless-steel canister in a strict vertical orientation underneath the AIDA chamber. One of the 10 mm inner diameter vertical sampling tubes of the chamber was connected to the SID-3 canister in a way that the tube end was located about 10 mm in front of the sensing area of the instrument. During the experiments air from the chamber was sampled through the sampling tube and through the instrument canister by applying a constant mass flow of 50 standard l/min resulting in a terminal particle speed of about 10 m/s at the sensing area of the instrument.



### 2.3 Retrieval scheme for deriving the optical constants of β-NAD

The iterative scheme for deriving the optical constants of β-NAD from the measured infrared extinction spectrum follows our approach recently developed for ammonium nitrate particles (Wagner et al., 2021). We briefly summarise it here with the adjustments that were necessary for the case of β-NAD. The flowchart of the retrieval procedure is shown in Fig. 3. The wavenumber-dependent optical constants are the real, $n(\tilde{v})$, and the imaginary part, $k(\tilde{v})$, of the complex refractive index $N(\tilde{v})$, i.e. $N(\tilde{v}) = n(\tilde{v}) + ik(\tilde{v})$. The two quantities are not independent, but are connected by the Kramers-Kronig

relationship, which in its subtractive form can be written as follows (Ahrenkiel, 1971; Milham et al., 1981; Segal-Rosenheimer and Linker, 2009):

$$n\left(\tilde{v}_k\right) = n\left(\tilde{v}_x\right) + \frac{2\left(\tilde{v}_k^2 - \tilde{v}_x^2\right)}{\pi} P \int_0^\infty \frac{k\left(\tilde{v}\right)\tilde{v}}{\left(\tilde{v}^2 - \tilde{v}_k^2\right)\left(\tilde{v}^2 - \tilde{v}_x^2\right)} d\tilde{v} \qquad (1)$$

$P$ is the notation for the Cauchy principal value of the integral. The real part of $N(\tilde{v})$ can thus be calculated for any given wavenumber $\tilde{v}_k$, provided that $k(\tilde{v})$ is known for a sufficiently large wavenumber range. The integral is evaluated around the

anchor point value $n(\tilde{v}_x)$, which is a known value for the real part of the complex refractive index at a certain wavenumber $\tilde{v}_x$. The basic idea of the retrieval method is to start with an initial guess of the $k(\tilde{v})$ spectrum, derive the corresponding $n(\tilde{v})$ spectrum with Eq. (1), and use these in an optical model together with the size distribution of the β-NAD particles to calculate the infrared extinction spectrum. An optimisation algorithm is then used to iteratively adjust the $k(\tilde{v})$ spectrum to minimise the root mean square deviation (RMSD) between the measured and the calculated extinction spectrum.


The initial guess for $k(\tilde{v})$ was derived from a literature spectrum. For this purpose, we digitised the spectrum of the annealed film from the NAD spectra collection of Tisdale et al. (1999) (their Fig. 4, trace b), which, similar to the spectrum of the annealed particles mentioned in the Introduction, clearly exhibits the spectral characteristics of β-NAD reported in Grothe et al. (2004). The spectrum was interpolated to the wavenumber grid of our measurement, which was provided at a digital

resolution of about 2 cm⁻¹, resulting from an approximate doubling of the size of the original interferogram due to zero-filling by a factor of two (Aroui et al., 2012). For thin film spectra, the absorbance is directly proportional to $k(\tilde{v}) \cdot \tilde{v}$. Hence, we were able to immediately derive $k(\tilde{v})$ with an estimate for the proportionality factor, i.e. the film thickness, and scale it such that the resulting $k(\tilde{v})$ spectrum had peak values similar to α-NAD in the wavenumber range of the O–H stretching modes (3500 – 3000 cm⁻¹) (Toon et al., 1994; Niedziela et al., 1998). The initial guess $k(\tilde{v})$ spectrum derived in this way from Tisdale et

al. (1999) extended to a wavenumber of about 500 cm⁻¹, whereas our infrared measurements had a low cut-off frequency of 800 cm⁻¹. The iterative adjustment of the $k(\tilde{v})$ spectrum was therefore limited to 800 cm⁻¹, but we always used the 800 – 500 cm⁻¹ range of the initial guess $k(\tilde{v})$ spectrum as a low-frequency extension when calculating the Kramers-Kronig integral to minimise truncation errors caused by a limited integration range (Segal-Rosenheimer and Linker, 2009). The Kramers-Kronig integration was performed using Maclaurin's formula method (Ohta and Ishida, 1988). The anchor point was set at 4000



cm$^{-1}$. All previously derived data sets of infrared refractive indices for α- and β-NAT (Toon et al., 1994; Richwine et al., 1995) and α-NAD (Toon et al., 1994; Niedziela et al., 1998) showed little variation in the real part of the complex refractive index at this wavenumber (1.40 – 1.41 for α- and β-NAT, 1.42 for α-NAD). Hence, we chose a value of $n(4000 \text{ cm}^{-1}) = 1.42$ for the anchor point.

Regarding the optical model, it was essential to go beyond Mie theory, which is computationally efficient but only applies to spherical particles, and explicitly consider aspherical particle habits. We already mentioned in the Introduction that the nucleated β-NAD particles are rather compact in shape and do not differ too much from a spherical geometry (a more detailed investigation is presented in Sect. 3.3). However, even small deviations from a spherical shape can notably affect the spectral habitus under certain resonance conditions ($n \approx 0$ and $k \approx \sqrt{2}$), where so-called Fröhlich, or surface modes appear in the particle

extinction spectrum (Bohren and Huffman, 1983; Clapp and Miller, 1993; Wagner et al., 2021). Averaging over a distribution of aspect ratios does not cancel the shape effect, i.e., does not reproduce the Mie solution. We therefore used the T-matrix method and modelled the β-NAD particles as randomly oriented spheroids (Mishchenko and Travis, 1998), employing the aspect ratio distribution derived from the analysis of the two-dimensional SID-3 scattering patterns presented above. The extinction spectrum of the particles was calculated by interpolation from look-up tables of the extinction efficiency on a three-

dimensional parameter space of $n$, $k$, and the size parameter $x_p$ ($x_p = \pi d_p / \lambda$; $d_p$: equal-volume sphere diameter, $\lambda$: wavelength) (Wagner et al., 2021). These look-up tables were computed for 5 different aspect ratios in the parameter space of $n$ between 0.4 and 4 (37 values with $\Delta n = 0.1$), $k$ between 0.0001 and 3.5 (39 values: 0.0001, 0.025, 0.05, 0.075, 0.1 – 3.5 with $\Delta k = 0.1$), and $x_p$ between 0.005 and 6 (31 values: 0.005, 0.01, 0.03, 0.05, 0.1, 0.2 – 4.0 with $\Delta x_p = 0.2$, 4.25, 4.5, 4.75, 5, 5.5, and 6), summing up to 44733 calculations for each aspect ratio. Both oblate ($\phi > 1$) and prolate ($\phi < 1$) particle shapes ($\phi = 0.5$, 0.67,

1, 1.5, and 2) were considered in the computations, onto which the aspect ratio distribution measured with the SID-3 was then projected to calculate the shape-averaged extinction spectrum of the β-NAD particles. It was assumed that the shape distribution is identical in all particle size bins.

        In addition to the optical constants and the shape distribution, the T-matrix calculation requires the particle size distribution as

a further input variable. While the number concentration of the volatile β-NAD particles was measured with an accuracy of ± 20% with the welas OPC in the cold isolating containment of the AIDA chamber, particle sizing by the instrument is less accurate for aspherical particle shapes. However, we have already mentioned above that, according to previous studies, the real index of refraction at non-absorbing wavenumbers > 4000 cm$^{-1}$ is very similar for all NAD and NAT polymorphs. We therefore used the α-NAD optical constants of Toon et al. (1994) between 6000 and 4000 cm$^{-1}$, i.e., in the range of the

extinction spectrum that is dominated by light scattering and does not contain any absorption bands, as an estimate for the β-NAD optical constants. We specified the size distribution as log-normal, and retrieved the best-fitted parameters for the mode width and the count median diameter of the β-NAD crystals (expressed as equal-volume sphere diameter) by minimising the



RMSD between the measured and the computed extinction spectrum, fixing the particle number concentration to the OPC measurement. A similar strategy has been successfully applied in previous infrared measurements to determine the growth of

supercooled $H_2SO_4/H_2O$ solution droplets during an expansion cooling experiment in the AIDA chamber (Wagner et al., 2008).

With the size- and shape distribution of the β-NAD crystals and the initial guess of the $k(\tilde{v})$ spectrum, we could then compute the size- and shape-averaged extinction spectrum as well as the RMSD with respect to the measured spectrum. In the optimisation step, which consisted of minimising the RMSD by iterating $k(\tilde{v})$, we first fitted the $k(\tilde{v})$ spectrum derived from

Tisdale et al. (1999) by a superposition of 20 gaussian modes, each defined by its amplitude, width, and peak location. This reduced the retrieval procedure to only 60 optimisation parameters and allowed a good approximation of the true $k(\tilde{v})$ spectrum in a reasonable amount of computer time. The downhill simplex method was used as the optimisation algorithm (Press et al., 1992). For the fine-tuning of the $k(\tilde{v})$ spectrum, we then included all 1601 individual wavenumber points between 4000 and 800 $cm^{-1}$ in the optimisation procedure. To improve and accelerate the convergence behaviour of the optimisation

algorithm at wavenumber ranges that contained only weak and spectrally broad absorption bands, we smoothed the $k(\tilde{v})$ spectrum in such ranges with a Savitzky-Golay filter (quadratic polynomial fit with 5 data points in the moving window) (Press et al., 1992; Wagner et al., 2021).

The procedure described above implies that the infrared extinction spectra recorded during the heterogeneous crystallisation

experiments can be interpreted to a good approximation as pure β-NAD spectra, i.e., that the inclusions of illite and $MgFeSiO_4$ particles in the β-NAD crystals have a negligible influence on the infrared spectral signatures. In Sect. 3.3, we will analyse the uncertainties of this approximation using simulations with a coated sphere optical mode.

# 3 Results and discussion

## 3.1 Overview: Homogeneous α-NAD and heterogeneous β-NAD nucleation experiments

In Fig. 4, we show selected time series of the most important AIDA measurement data in the course of a homogeneous (left part) and a heterogeneous NAD nucleation experiment (right part, illite as refractory material). Time zero denotes the start of the injection of the mixed $HNO_3$ and $H_2O$ carrier gases without (left) and with added solid seed aerosol particles (right). The duration of the injection is indicated by the magenta-coloured frame. Panels (a) show the AIDA pressure (black line) and the mean AIDA gas temperature (red line). The time evolution of the relative humidity with respect to ice ($RH_{ice}$, brown line) and

the saturation ratio with respect to NAD ($S_{NAD}$, green line) are shown in panels (b). To derive $S_{NAD}$, we first computed the activity product of the ions ($H^+$, $NO_3^-$) and the solvent ($H_2O$) in the liquid phase with the E-AIM model for the temperature and relative humidity conditions prevalent in the AIDA chamber, excluding the formation of solid phases (Clegg et al., 1998; Massucci et al., 1999). The respective activity product in a solution saturated with respect to solid NAD was computed





according to the temperature-dependent function given in Eq. (A9) of Massucci et al. (1999). The ratio of the two activity
products then yielded $S_{NAD}$. When the green solid lines in panels (b) turn into dotted lines, $RH_{liq}$ has fallen below the 10%
input range of the E-AIM model and the further evolution of $S_{NAD}$ is calculated using this fixed $RH_{liq}$ lower limit. $S_{NAD}$ thus
represents a lower limit in these periods as $RH_{liq}$ decreased even further. The blue dots in panels (c) symbolise the single
particle measurements with the welas OPC, where each dot represents a particle that has been assigned a specific optical
diameter according to its scattering intensity. The vertical arrows indicate the times of the FTIR spectra, which are shown and
analysed later in Figs. 5 and 6. The number concentration of solid NAD particles nucleated during the experiments is shown
as the orange line in panels (d). While $HNO_3/H_2O$ solution droplets without and with embedded illite particles were still
present, we defined an optical threshold size of 1 μm and counted all particles above that size as either α- or β-NAD crystals.
During this period, the reported number concentrations of NAD particles are slightly underestimated due to the small fraction
of crystals whose optical diameters overlapped with those of the solution droplets. For the heterogeneous nucleation
experiment, we also included the records of the number concentration of co-added illite particles measured with the CPC3010
through a heated sampling line (pink line). Finally, panels (e) show the trace of the near-backscattering linear depolarisation
ratio from the SIMONE instrument (δ, grey line).

The initial composition and mass concentration of the $HNO_3/H_2O$ solution droplets in the homogeneous nucleation experiment,
as determined from the FTIR spectra, were about 50 wt% $HNO_3$ and 2.7 mg/m³, respectively. The fact that the FTIR retrieval
is only sensitive to the mass concentration, but not to the exact shape of the size distribution of the aerosol particles, is a
consequence of the submicron size of the droplets, i.e., the size parameter at infrared wavelengths is still close to the Rayleigh
scattering regime (Bohren and Huffman, 1983). However, as shown in Fig. 6 of Stetzer et al. (2006), we can use the tail of the
size distribution, quantitatively measured with the welas OPC at diameters above 0.7 μm, to constrain the individual parameters
of the assumed log-normal size distribution of the added $HNO_3/H_2O$ solution droplets. Such an analysis yielded a droplet
number concentration of about 25000 cm⁻³, a mode width $\sigma_g$ of about 1.2, and a count median diameter of about 0.5 μm.
During the injection of the carrier gases and the associated nucleation of the $HNO_3/H_2O$ solution droplets, $RH_{ice}$ temporarily
increased to about 60%. In the course of the following observation period of 4.5 h, the $HNO_3/H_2O$ solution droplets slowly
evaporated with different rates for the deposition of water and nitric acid vapours to the cold chamber walls (Möhler et al.,
2006a; Stetzer et al., 2006). As a result, the relative humidity decreased and the nitric acid content in the solution droplets
increased to about 63 wt% by the end of the observation period. About 5000 s after the start of the injection, when $S_{NAD}$ had
risen above a value of 10, we observed the nucleation of the first larger NAD crystals in the records of the welas OPC (Fig.
4c, left part). The fact that the NAD crystals appeared with larger diameters in the OPC recordings is initially due to the
typically higher scattering efficiency of non-spherical particles compared to equal-volume spheres at scattering angles around
90° used in the OPC's detection system, so that they are assigned a larger optical diameter. However, in the course of the
observation period, the nucleated NAD crystals also actually grew in size at the expense of the $HNO_3/H_2O$ solution droplets,





due to the higher partial pressures over the liquid phase. After about 15000 s, the droplet mode had disappeared, leaving behind a population of pure NAD crystals with a number concentration of about 3 cm$^{-3}$, which led to an increase in the near-backscattering depolarisation ratio to about 11% (Fig. 4e, left part). We will show in Sect. 3.2 that the recorded infrared
spectrum of the crystalline particles was consistent with the α-NAD modification and exhibited the same spectral changes indicative of strongly aspherical particle habits as discussed in Wagner et al. (2005a). However, before we look at the shape of the homogeneously nucleated α-NAD crystals, we would like to describe the general observations made during the heterogeneous NAD nucleation experiment shown in the right-hand part of Fig. 4. Note that the corresponding panels (a-e) of Fig. 4 are vertically scaled identically to allow an immediate comparison of the two types of experiments.

The heterogeneous nucleation experiment was carried out in the same way as its homogeneous counterpart at a constant temperature of approximately 190 K. Using the same analysis as described above for the homogeneous nucleation experiment, we deduced a number concentration of approximately 20000 cm$^{-3}$ for the generated HNO$_3$/H$_2$O solution droplets. With a mode width of 1.2 and a count median diameter of 0.5 μm, the aerosol mass concentration was approximately 2.2 mg/m$^3$. The initial
HNO$_3$ content of the droplets was about 53 wt%. The number concentration of illite particles co-added with the HNO$_3$ – H$_2$O vapour mixture reached a maximum value of about 100 cm$^{-3}$ (Fig. 4d, right part, pink line). Thus, apart from a small subset of internally-mixed illite/HNO$_3$/H$_2$O particles, the injection also produced a large reservoir of purely liquid nitric acid solution droplets. Already during the injection and in the initial observation period, when S$_{NAD}$ was still below 10, we observed a considerable amount of larger, crystalline particles in the OPC records, documenting a much faster nucleation compared to the
homogeneous run shown next to it, which can only be explained by the presence of the illite particles acting as immersion nuclei for the crystallisation of the HNO$_3$/H$_2$O solution droplets. More than an order of magnitude higher numbers of NAD crystals were formed in the heterogeneous nucleation experiment than in the homogeneous run, and more than 80% of the co-added illite particles acted as heterogeneous nuclei for the crystallization of NAD (ratio of the orange line to the pink line in the right part of Fig. 4d). The large reservoir of HNO$_3$/H$_2$O solution droplets already disappeared shortly after 10000 s, leaving
behind an ensemble of NAD crystals that caused a much higher depolarisation ratio of 36% compared to the homogeneous crystallisation experiment.

Figure 5 shows a compilation of FTIR spectra illustrating that the heterogeneous crystallisation experiments not only accelerated the formation of solid particles, but also led to the formation of a different phase of the nucleated NAD particles, namely β-NAD. The infrared spectrum recorded at the end of the heterogeneous nucleation experiment with illite particles
(Fig. 4c, right part, t = 11000 s) is shown as trace b. A very similar spectrum was obtained in a different experiment at 190 K when using MgFeSiO$_4$ as refractory material to induce the crystallisation of the HNO$_3$/H$_2$O solution droplets (trace c). We first discuss these two spectra in comparison with a reference spectrum of compactly shaped α-NAD particles from our previous work (Wagner et al., 2005a), which were produced by shock freezing a gas mixture of nitric acid and water (trace a). The



spectral differences between spectra b and c compared to spectrum a show the same features discussed by Grothe et al. (2004) to distinguish between β- and α-NAD. In the left part of Fig. 5, which shows the full wavenumber range from 4000 to 800 cm$^{-1}$, we have marked with black vertical lines the two distinct maxima of α-NAD at 3486 and 3254 cm$^{-1}$ in the range of the O-H stretching modes. In contrast, the extinction spectra of the β-NAD particles reveal three distinct peaks at 3464, 3362, and 3138 cm$^{-1}$. Due to the contribution of light scattering, the peak positions of these extinction bands are shifted by 20 – 30 cm$^{-1}$

to lower wavenumbers in comparison with the IR absorption bands of β-NAD tabulated in Grothe et al. (2004), where the infrared spectrum of a β-NAD film deposited on a gold-platted sample support was measured in the specular reflectance mode. A more straightforward analysis of the band positions is presented later in Sect. 3.3, when we can use the retrieved spectrum of the imaginary part of the complex refractive index of β-NAD to compare the location of the absorption peaks. In the wavenumber range between 3000 and 2000 cm$^{-1}$, which is dominated by symmetric and asymmetric stretching modes of the

$H_3O^+$ ion, and between 2000 and 1700 cm$^{-1}$, which contains the asymmetric deformation mode of $H_3O^+$, $\nu_4(H_3O^+)$, the spectral differences between β- and α-NAD are rather subtle and manifest themselves in only minor changes of the band shapes. More distinct changes are again clearly visible at wavenumbers between 1600 and 800 cm$^{-1}$, which are shown in an expanded view in the right part of Fig. 5. A distinct marker to distinguish between the two NAD polymorphs is the location of the $\nu_1(NO_3^-)$ mode, which experiences a considerable blue shift from about 1026 cm$^{-1}$ for α-NAD (vertical black line the right part of Fig.

5) to about 1038 cm$^{-1}$ in the case of β-NAD. Characteristic of β-NAD are also two bands at lower wavenumbers, as indicated by the vertical blue lines in the right part of Fig. 5, namely the so far unassigned mode at about 852 cm$^{-1}$ and the intense $\nu_2(NO_3^-)$ mode at about 812 cm$^{-1}$ (Grothe et al., 2004).

    Clear spectral differences between β- and α-NAD also appear at wavenumbers between 1500 and 1100 cm$^{-1}$, where the

observed extinction signatures are due to the intense $\nu_3(NO_3^-)$ mode and the somewhat weaker $\nu_2(H_3O^+)$ mode (Fernandez et al., 2003). However, as we will see in Sects. 3.2 and 3.3, the spectral habitus in this wavenumber range is also very sensitive to the size and shape of the NAD crystals, which makes this not the best criterion for distinguishing between the two polymorphs. Figure 5 contains a third exemplary extinction spectrum of β-NAD crystals that we recorded at the end of a heterogeneous crystallisation experiment with the illite particles conducted at a higher temperature of 200 K (trace d). It bears

all the characteristic markers of β-NAD discussed in the preceding paragraph, but shows a more pronounced shoulder at 1150 cm$^{-1}$, which we will discuss as an effect of particle size in Sect. 3.3. The recordings b, c, and d are, to our knowledge, the first infrared spectra of aerosol particles where direct crystallisation of β-NAD from the supercooled liquid phase has been observed. While we have compiled in Fig. 5 only the spectra after full crystallisation (i.e., after the disappearance of the droplet mode), we also investigated the spectral habitus at earlier stages of the heterogeneous crystallisation experiments, where we

subtracted the signature of the $HNO_3/H_2O$ solution droplets to deduce the spectrum of the freshly crystallised particles. In this analysis, we also observed all characteristic markers of β-NAD. Furthermore, as we will see in Sect. 3.2, homogeneously nucleated α-NAD crystals produced very peculiar two-dimensional scattering patterns in the SID-3 images, whereas no such



scattering patterns were ever detected in the course of the heterogeneous crystallisation experiments. So either the α-NAD phase did not form at all, or was very rapidly transformed into β-NAD. Regardless of which case applies, it can be summarised

that heterogeneous crystallisation aided by the surface of refractory materials (e.g. of the meteoric smoke or mineral dust type) leads to the efficient formation of the more stable, three-dimensionally ordered β-NAD phase. In contrast, the α-NAD polymorph, whose structure matches more the short-range order of the liquid phase, crystallises from $HNO_3/H_2O$ solution droplets without solid inclusions, in accordance with Ostwald's step rule (Grothe et al., 2004). The transformation into β-NAD can then only be induced by warming (Tisdale et al., 1999; Grothe et al., 2004; Grothe et al., 2008). Spectrum e in Fig. 5 shows

the annealed particle spectrum from Tisdale et al. (1999) (Fig. 4 therein, trace E). The smaller particle size in that spectrum compared to our experiments (spectra b – d) reduces the scattering contribution and alters the signature of the extinction bands, but the key features typical of β-NAD are clearly visible, so this is presumably the first record of β-NAD, even if it was not identified as such at that time.

**3.2 Particle shape of homogeneously nucleated α-NAD crystals**

In Fig. 6a we show our earlier FTIR measurements, which led us to believe that strongly aspherical α-NAD crystals can form when supercooled $HNO_3/H_2O$ solution droplets freeze and slowly grow in a supersaturated environment (Wagner et al., 2005a). In the wavenumber range between 1600 and 800 $cm^{-1}$, we observed a strong modulation of the infrared spectral habitus between near-spherical α-NAD particles obtained by shock-freezing a gaseous mixture of nitric acid and water (exp. A from Wagner et al. (2005a), blue line) and that of α-NAD crystals generated by homogeneous nucleation from slowly evaporating

supercooled $HNO_3/H_2O$ solution droplets at 193 K over a time scale of several hours (exp. B1 from Wagner et al. (2005a), red line). As shown in the accompanying analysis, aspect ratios ≤ 0.2 or ≥ 5 were required to adequately describe the band shape of spectrum B1 (Wagner et al., 2005a). However, the spectral analysis did not provide an unambiguous indication of whether the crystals were of prolate or oblate shape. The black coloured infrared spectrum in Fig. 6a was recorded at the end of the homogeneous nucleation experiment from this study (see left part of Fig. 4), and its habit is consistent with that of the earlier

experiment B1. A collection of representative two-dimensional SID-3 scattering images of the nucleated α-NAD crystals from this experiment is shown in Fig. 6b. The images reveal two-fold symmetrical scattering patterns with large differences in the number of diffraction maxima along the two symmetry axes, which is indicative of highly elongated, needle-like particle shapes. The quantitative analysis of the scattering patterns in terms of the number ratio of diffraction maxima along the two symmetry axes gave a median inverse aspect ratio of 5 ± 2, with maximum values up to 10. These values are in very good

agreement with the observation of prolate α-NAD particles of similar aspect ratio on the cryo-stage of an ESEM (Grothe et al., 2008).

Our measurements with the SIMONE light scattering instrument indicate that elongated nitric acid hydrate crystals only cause a relatively low near-backscattering linear depolarisation ratio of about 11% (Fig. 4e, left part). In T-matrix computations





where non-spherical nitric acid hydrates were modelled as randomly oriented spheroids or cylinders, moderate aspect ratios

between 1.2 and 2 (both prolate and oblate shapes were considered) gave linear depolarisation ratios in the range of about 25

– 40% for particles with equal-volume sphere diameters between 2 and 4 μm (Liu and Mishchenko, 2001). In fact, this range

agrees with the near-backscattering linear depolarisation ratios that we observed in the heterogeneous nucleation experiments

where β-NAD particles were formed (see Fig. 4e, right part). In Sect. 3.3, we will use the SID-3 images to prove that the

nucleated β-NAD crystals did indeed have a rather compact particle shape within that range of aspect ratios. The much lower

depolarisation ratio of the α-NAD crystals compared to β-NAD is due to the peculiar scattering properties of particles with

aspect ratios that greatly deviate from unity. Using ice crystals as an example, Zakharova and Mishchenko (2000) have shown

that wavelength-sized needle- and plate-like particles, modelled as spheroids with aspect ratios of 0.05 and 20, cause much

less backscattering linear depolarisation than surface-equivalent particles with moderate aspect ratios of 0.5 and 2. The large

variability of the depolarisation ratio as a function of particle shape, as seen in Fig. 4e, could impact the accuracy by which the

amount of nitric acid hydrates in mixtures with STS droplets can be retrieved from lidar measurements (Pitts et al., 2009; Pitts

et al., 2018).

### 3.3 Particle shape and infrared optical constants of β-NAD particles

In the previous section, we already anticipated that one would expect a rather compact particle shape for the β-NAD crystals

due to the magnitude of the measured near-backscattering linear depolarisation ratio. In Fig. 7 we now show a selection of

SID-3 scattering patterns from two heterogeneous crystallisation experiments, the left part showing the measurements from

the experiment with illite as refractory seeding material (as shown in the right part of Fig. 4), and the right part from a similarly

performed experiment with $MgFeSiO_4$ as solid inclusion. None of the elongated scattering patterns typical of homogeneously

nucleated α-NAD crystals can be seen in these images. As already discussed in connection with Fig. 4, heterogeneous

nucleation proceeded very rapidly and at low $S_{NAD}$; hence, we conclude from the lack of elongated scattering patterns that in

fact the entire aerosol population had crystallised heterogeneously and that only β-NAD was formed. The large proportion of

$HNO_3/H_2O$ solution droplets without solid inclusion therefore only served as a reservoir of condensable material for the growth

of the nucleated β-NAD crystals and was consumed in the process. Figure 5 reveals that the two infrared spectra of the β-NAD

crystals from the illite and the $MgFeSiO_4$ experiment are very similar (spectra b and c), as are the two sets of SID-3 scattering

patterns shown in Fig. 7. They feature a spatially more uniform and thus less directional illumination of the ICCD sensor

compared to the highly directional patterns that are characteristic for α-NAD particles. For β-NAD, the analysis in terms of

diffraction maxima along the two symmetry axes yielded a median value of 1.5 ± 0.5. Less than 5 % of the images had an axial

ratio greater than two. For these compactly shaped β-NAD crystals, it is not evident from the scattering patterns whether the

particles have a prolate or an oblate shape, so that the value given above can be assigned either to oblate particles with an

aspect ratio of 1.5 : 1 or to prolate particles with an aspect ratio of 1 : 1.5. Within the range from 1 to 2, the histogram of axial

ratios was relatively evenly distributed, so we computed an averaged look-up table of the extinction efficiencies for an





equivalent mixture of all five considered aspect ratios, that is $\phi = 0.5, 0.67, 1, 1.5$, and 2, to retrieve the optical constants of β-NAD.

Figure 8 shows the newly derived data set of optical constants of β-NAD. We have highlighted four regions of the $k$ spectrum whose absorption signatures are discussed in the following. In Table 1 we have listed the location of the infrared absorption bands in comparison with those assigned by Grothe et al. (2004). In regions (I), (II), and (IV), there is a good match between the peak positions from both studies. Our $k$ spectrum shows the splitting of the $\nu_{1,3}(H_2O)$ vibration into three bands, which is characteristic for the β-NAD modification (region I). Three spectrally broad absorption bands due to the $\nu_{1,3}(H_3O^+)$ and
$\nu_4(H_3O^+)$ modes cover the wavenumber range from 2900 to 1600 cm$^{-1}$ (region II). In region (IV), in very good agreement with Grothe et al. (2004), we find two narrow peaks at 1038 and 812 cm$^{-1}$ attributed to the $\nu_1(NO_3^-)$ and $\nu_2(NO_3^-)$ modes, respectively, and another band at 852 cm$^{-1}$ yet unassigned. In the range between about 1500 and 1100 cm$^{-1}$ with the $\nu_3(NO_3^-)$ and $\nu_2(H_3O^+)$ modes (region III), we observe a particularly strong absorption band at 1228 cm$^{-1}$. This strong resonance signal is also clearly visible in the spectra of the annealed film and annealed particles of Tisdale et al. (1999), but is much less
pronounced and shifted in wavenumber in the vibrational spectra presented in Grothe et al. (2004). While the thin films in Tisdale et al. (1999) were studied by single-pass transmission infrared spectroscopy on a silicon wafer, Grothe et al. (2004) used a reflective gold-platted support to deposit the samples. Using the example of a crystalline α-NAT film, it was shown that the specific orientation of the molecules on an underlying gold surface can either enhance or suppress the intensities of the infrared bands in reflection–absorption spectroscopy compared to transmission measurements, depending on the orientation
of the transition dipole moment (Koch et al., 1996). For example, the $\nu_3(NO_3^-)$ mode of crystalline α-NAT was considerably less infrared active when the sample was deposited on a gold substrate compared to a transmission measurement on a silicon wafer (Koch et al., 1996; Tisdale et al., 1997). We assume that such orientation effects might contribute to the partial discrepancies between the $k$ spectrum we determined for β-NAD and the infrared measurement by Grothe et al. (2004).

In Fig. 9a, we show the measured spectrum of β-NAD from the heterogeneous nucleation experiment with illite at 190 K together with the calculated spectrum after the convergence of the optimisation algorithm, which yielded the optical constants displayed in Fig. 8. We then used this new $n$ and $k$ data set to fit the β-NAD spectrum from the crystallisation experiment conducted at 200 K where we observed the pronounced shoulder at 1150 cm$^{-1}$ in the spectral habitus (see Fig. 5, trace d). In this fit we used the same shape distribution and just varied the log-normal parameters of the size distribution of the β-NAD
particles until the best agreement between the measured and the calculated spectrum was obtained. Figure 9b shows that the shoulder-like signature can be reproduced very accurately. The difference in the habitus compared to the spectra shown in Fig. 9a is simply related to particle size. While the particle diameter in the experiment at 190 K corresponded to about 2 µm, a value of about 2.5 µm was retrieved for the experiment at 200 K. We also attempted to reproduce the annealed particle extinction spectrum of Tisdale et al. (1999), which has a much lower scattering contribution because the particles were in the





submicron size range. Our best-fitted spectrum, as shown in Fig. 9c, agrees quite well with the measured spectrum, in particular the strong extinction band between 1400 and 1100 cm$^{-1}$ is accurately reproduced. Minor spectral inaccuracies must be acknowledged due to the necessary digitisation of the spectrum from the figure in the publication. Other, more physical reasons for the remaining differences between the measured and calculated spectra could be the temperature dependence of the optical constants, a not complete conversion from the α- to the β-modification during annealing, and the simplified assumption of a

spheroidal particle shape and an identical shape distribution in all size bins in the optical model. Nevertheless, the three fitting examples from Fig. 9 highlight that we can accurately reproduce the β-NAD extinction signatures for different particle sizes, which is an indication of the robustness of the derived $n$ and $k$ data set.

In the quantitative assessment of the uncertainty of our new data set, we consider two important aspects, namely the influence

of the particle shape on the extinction signatures and the treatment of the β-NAD particles as pure crystals with neglect of refractory inclusions. Regarding the first point, we show in Fig. 10 the results of a series of T-matrix computations with the retrieved $n$ and $k$ data set where we varied the aspect ratio of the β-NAD particles. Panel (a) covers the entire wavenumber range from 4000 to 800 cm$^{-1}$ whereas panel (b) contains an expanded view from 1600 to 800 cm$^{-1}$. The black spectrum is the calculation for the equivalent mixture of the five aspect ratios ($\phi = 0.5$, 0.67, 1, 1.5, and 2) as used in the retrieval of the optical

constants. This spectrum is equal to the best-fitted spectrum shown in Fig. 9a for the heterogeneous nucleation experiment with illite at 190 K, where the β-NAD particles grew to a size of 2 μm. For the other three calculations, the particle size distribution was kept constant, but individual aspect ratios of $\phi = 1$, 2, and 0.5 were used instead of the shape mixture. The extent of the shape-dependent variations in the extinction spectrum of β-NAD was similar to that observed in our preceding study with crystalline ammonium sulphate (Wagner et al., 2021). While the influence of particle shape is negligible in spectral

regions with small or moderate $k$ values ($k < 0.5$), the strong absorption centred at 1228 cm$^{-1}$ corresponds to a situation where high values for the imaginary index ($k \gg 1$) are accompanied by anomalous dispersion signals with large amplitude in the $n$ spectrum. Such conditions favour the occurrence of Fröhlich or surface modes that are strongly dependent on the particle shape (Bohren and Huffman, 1983; Clapp and Miller, 1993). The maximum of the extinction signal is shifted with respect to the centre of the $k$ absorption peak and is located at 1244 cm$^{-1}$ in the calculation for the shape mixture. Pure prolate ($\phi = 0.5$) and

oblate ($\phi = 2$) particles cause a small shift in the position of this 1244 cm$^{-1}$ band compared to the shape mixture and change its intensity in opposite directions. The sphere calculation ($\phi = 1$) does not represent an average of the two spheroidal shapes, but agrees better with the computation for $\phi = 2$ in terms of band intensity. If the individual shapes $\phi = 1$, 2, and 0.5 and not the shape mixture were used to determine the optical constants, the spectral differences in Fig. 10b would be reflected in changes in the retrieved $n$ and $k$ data set. In Fig. 10c we show the result of such an analysis in which we repeated the retrieval procedure

but used $\phi = 1$, 2, and 0.5 to represent the shape of the β-NAD crystals. We only show the range of the $k$ spectrum from 1600 to 800 cm$^{-1}$ because the changes in other spectral regions were insignificant. We observe maximum variations of ± 12% for the peak intensities of both the 1228 cm$^{-1}$ band and the $\nu_1(NO_3^-)$ mode at 1038 cm$^{-1}$ in comparison with the retrieval result for





the shape mixture. The 1228 cm⁻¹ mode is also subject to band shifts of ± 10 cm⁻¹, while the position of the $\nu_1(NO_3^-)$ mode does not depend on the particle shape. We consider these variations as the error regime of the retrieval results in terms of shape

dependency. The $n$ and $k$ spectra for the equivalent mixture of five shapes are in the middle of the range of results for individual particle shapes, and are considered by us to be the most reliable solution. Therefore, we offer this dataset as a download via the link provided under Data availability.

As mentioned in Sect. 2.3, our retrieval procedure implies that the infrared extinction spectra recorded during the

heterogeneous crystallisation experiments can be interpreted to a good approximation as pure β-NAD spectra, i.e., that the inclusions of the illite and $MgFeSiO_4$ particles in the β-NAD crystals can be neglected. Independent measurements of the number size distribution of the two refractory particle types after dispersion with the rotating brush generator are shown in Fig. 11a. The count median diameters are in the range between 0.20 and 0.25 μm. More than 95% of the particles had sizes ≤ 0.50 μm. As explained above, β-NAD particles that crystallised on these seed aerosol particles, when mixed internally with

aqueous nitric acid, were able to grow at the expense of the reservoir of pure aqueous $HNO_3/H_2O$ solution droplets, reaching a median size of ≥ 2 μm. In the following, we discuss the results of a Mie simulation in which we modelled the composite particles as coated spheres using the BHCOAT subroutine from Bohren and Huffman (1983). First, we computed the wavenumber-dependent extinction cross sections, $C_{ext}$, of log-normally distributed, pure β-NAD spheres with a count median diameter of 2 μm and a mode width, $\sigma$, of 1.2. Then, we recalculated the spectrum with all β-NAD crystals containing a

monodisperse, spherical refractory inclusion of diameters varying between 0.20 and 0.50 μm. The inclusions replaced the inner part of the β-NAD crystals, leaving the overall particle diameter unchanged. For the Mie computations, we used our retrieved refractive indices for the β-modification to model the NAD part of the composite particles, while for the refractory inclusions we used optical constants for meteoric dust (Shettle and Volz, 1976). Selected results of our simulations are shown in Figs. 11b and 11c for the entire wavenumber range between 4000 and 800 cm⁻¹ and for the expanded region between 1600 and 800

cm⁻¹, respectively. With a diameter of 0.25 μm for the refractory inclusion, there is hardly any change compared to the extinction spectrum of the pure β-NAD particles. A slight shift in peak position and a 7% decrease in peak intensity of the strong extinction band at 1244 cm⁻¹ is observed when the diameter of the inclusion is increased to 0.50 μm. As noted above, there is a small fraction of seed aerosol particles with diameters larger than 0.50 μm that, if included in the composite particles, would trigger even larger changes in the extinction signatures compared to a pure β-NAD crystal than shown for the calculation

with an inclusion size of 0.50 μm. However, we have emphasised in Sect. 3.1 that a very large proportion > 80% of the seed aerosol particles has acted as nuclei for the crystallisation of β-NAD. Therefore, crystallisation is not preferentially induced by the small number fraction of particles > 0.50 μm, so that the largest fraction of β-NAD crystals still forms on smaller seed aerosol particles, whose influence on the extinction spectrum is very small. Therefore, we consider the computation for an inclusion with a diameter of 0.50 μm as a conservative upper estimate for the changes in the extinction signatures due to the



simplified treatment of the composite particles as pure β-NAD crystals in the retrieval scheme. The uncertainty is thus similar to that discussed above in relation to the influence of particle shape.

**4 Conclusions and outlook**

We have derived the first data set of infrared optical constants for the β-modification of nitric acid dihydrate. β-NAD is considered to be the system's high-temperature phase, because it was first observed and structurally characterised by X-ray

diffraction when the low-temperature α-NAD modification was heated above 210 K (Lebrun et al., 2001a). However, we have shown in this work that β-NAD particles were also efficiently formed during isothermal crystallisation experiments at 190 K, when the pre-added $HNO_3/H_2O$ solution droplets contained a solid inclusion of illite or $MgFeSiO_4$. Apparently, the surface of the embedded refractory materials supported the development of the three-dimensional network of H-bonds typical of β-NAD. Nitric acid solution droplets that did not contain a solid inclusion crystallised more slowly into the α-NAD modification, whose

structure better matches the short-range order of the liquid phase (Grothe et al., 2004). The two-dimensional scattering patterns measured with the SID-3 instrument revealed that the α-NAD particles predominantly grew into needle-like crystals, while the β-NAD particles had a more compact shape, which is consistent with the ESEM images of Grothe et al. (2008).

As outlined in the introduction, the potential ice-free formation pathway of nitric acid hydrates via heterogeneous nucleation

on particles of meteoric origin has inspired a number of laboratory studies, including recent work parameterising the nucleation activities of a variety of materials as active sites per unit surface area as a function of $S_{NAT}$ (James et al., 2018; James et al., 2022). Remarkably, all AIDA crystallisation experiments performed so far with binary $HNO_3/H_2O$ solution droplets (with or without solid inclusions) resulted in metastable nitric acid hydrates, i.e., either β- or α-NAD. In satellite- and aircraft-based infrared measurements, however, only β-NAT has been clearly identified, e.g. by its signature at 820 cm$^{-1}$ (Höpfner et al.,

2006; Woiwode et al., 2016; Woiwode et al., 2019; Lecours et al., 2022; Lecours et al., 2023). The new data set of optical constants for β-NAD clearly does not question this assignment, since the $v_2(NO_3^-)$ mode in β-NAD is located at 812 cm$^{-1}$. One could therefore ask about the missing link between the AIDA laboratory experiments (formation of α,β-NAD) and the atmospheric observations (detection of β-NAT). It was shown that β-NAD begins to decompose into NAM and β-NAT at a temperature of about 220 K (Grothe et al., 2004; Grothe et al., 2008). Therefore, it cannot be ruled out that such a transformation

also occurs at lower temperatures on time scales longer than the observation time in AIDA, which is only a few hours and limited by sedimentation of the large, micron-sized crystals. But are there scenarios in which β-NAT particles would crystallise directly? As mentioned before, an important aspect controlling the type and distribution of solid phases in the $HNO_3/H_2O$ system during crystallisation is the stoichiometry of the underlying liquid phase (Tizek et al., 2002; Tizek et al., 2004; Grothe et al., 2008). In our experiments, we started with a rather high nitric acid concentration of about 50 wt% in the $HNO_3/H_2O$

solution droplets, which is already at the upper end of $HNO_3$ concentrations reported for STS droplets (Meilinger et al., 1995),



and these droplets were further concentrated during crystallisation. Moreover, we have neglected the third component of the STS solution droplets, i.e., sulphuric acid, which could also influence the crystallisation behaviour.

A more atmospherically relevant experiment would therefore be to study the crystallisation of STS droplets containing an
inclusion of meteoric material. Previously, we have shown that we can simulate the uptake of nitric acid in sulphuric acid solution droplets to form pure STS particles at temperatures below 195 K in the AIDA chamber (Wagner et al., 2005b). In this study, $N_2O_5$ formed by the gas phase reaction between $NO_2$ and $O_3$ was hydrolysed on pre-added $H_2SO_4/H_2O$ solution droplets to form STS aerosol particles with a final composition of about 45 wt% $HNO_3$, 5 wt% $H_2SO_4$, and 50 wt% $H_2O$. For the above purpose, we would like to modify the procedure in a future experiment as follows. In the first step, we would inject the solid
particles that serve as a proxy for meteoric ablation material. In the second step, we would coat these particles in situ at low temperature with sulphuric acid, which is formed by the oxidation of sulphur dioxide with hydroxyl radicals (Bertozzi, 2021). The latter could be produced by the gas phase reaction between ozone and tetramethylethylene. Hydrolysis of $N_2O_5$ on the aqueous sulphuric acid coating layer would finally yield the desired STS droplets with a meteoric inclusion, whose crystallisation behaviour could then be studied. Another promising experiment for observing direct nucleation of β-NAT in
AIDA would be to adjust the environmental conditions (temperature and relative humidity) such that $S_{NAD} < 1$, and to choose a material from the studies of James et al. (2018, 2022) where high nucleation activity was observed for such conditions, so that the formation of β-NAD should be excluded. In a third type of experiment, the homogeneous freezing of pure STS solutions droplets at $T < 188$ K could be investigated. As the experiments of Weiss et al. (2016) have shown, α-NAT could easily form on the surface of the ice crystals and convert to β-NAT at $T > 195$ K.


Although it is tempting to look for direct NAT crystallisation pathways in future laboratory studies, our current experiments once again highlight how effectively metastable NAD phases can be formed due to a lower free-energy barrier for nucleation compared to NAT (Worsnop et al., 1993; Salcedo et al., 2001). Atmospheric observations should therefore continue to look for metastable phases, and as far as infrared measurements are concerned, our study has filled a gap by providing the necessary
optical constants for the β-NAD modification.

**Data availability**

Upon manuscript acceptance, we will archive the new refractive index data set for β-NAD derived in this work in the KITopen repository, the central publication platform for KIT (Karlsruhe Institute of Technology) scientists (Open Access, contact: KITopen@bibliothek.kit.edu), and assign them a citable persistent identifier (DOI).



## Author contributions

MS, RWe, and JMCP conceptualised and supervised the project. MS, RWa, ADJ, VLF, HS, OM, and RWe conducted the experiments at the AIDA chamber. ADJ and VLF synthesised the $MgFeSiO_4$ sample. RWa and MS analysed the data. RWa wrote the original manuscript draft with input from MS and RWe. All authors contributed to review and editing.

## Competing interests

The authors declare that they have no conflict of interest.

## Acknowledgments

We gratefully acknowledge the continuous support by all members of the Engineering and Infrastructure group of IMK-AAF, in particular by Olga Dombrowski, Rainer Buschbacher, Tomasz Chudy, Steffen Vogt, and Georg Scheurig. This work has been funded by the Helmholtz-Gemeinschaft Deutscher Forschungszentren as part of the program "Atmosphere and Climate". ADJ, BJM and JMCP were funded by the European Research Council (project number 291332 - CODITA) and VLF was funded by the Leverhulme Trust (F/00 122/BB – PETALS).

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





**Table 1: Assignment of infrared absorption bands (cm$^{-1}$, "sh": shoulder) of β-NAD by Grothe et al. (2004) in comparison with the peak positions from the *k* spectrum retrieved in this study (Fig. 8)**

|  | Grothe et al. (2004) | This work |
|---|---|---|
| $\nu_{1,3}(H_2O)$ | 3490 | 3482 |
|  | 3392 | 3382 |
|  | 3175 | 3150 |
| $\nu_{1,3}(H_3O^+)$ | 2717 | 2706 |
|  | 2273 | 2302 |
| $\nu_4(H_3O^+)$ | 1860 | 1772 |
| $\nu_3(NO_3^-)$ | 1460 | 1446 (sh) |
|  | 1430 | 1412 |
|  | (1336) | 1228 |
| $\nu_2(H_3O^+)$ | 1144 | 1150 (sh) |
| $\nu_1(NO_3^-)$ | 1040 | 1038 |
| ? | 855 | 852 |
| $\nu_2(NO_3^-)$ | 811 | 812 |



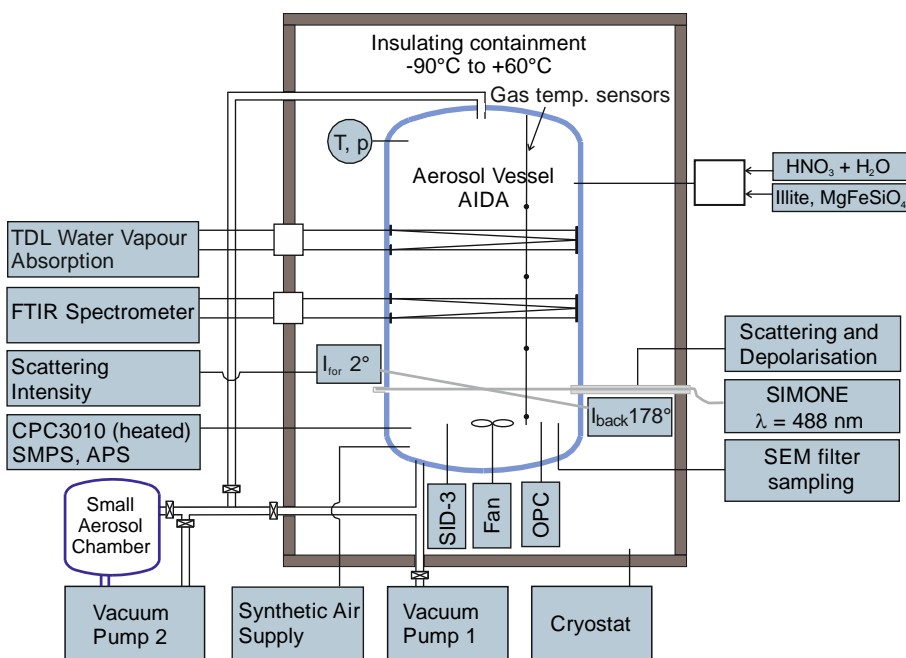


**Figure 1: Schematic diagram of the instrumentation of the AIDA aerosol and cloud chamber facility. The instruments (and their respective abbreviations) used in this work are described in more detail in Sects. 2.1 and 2.2.**






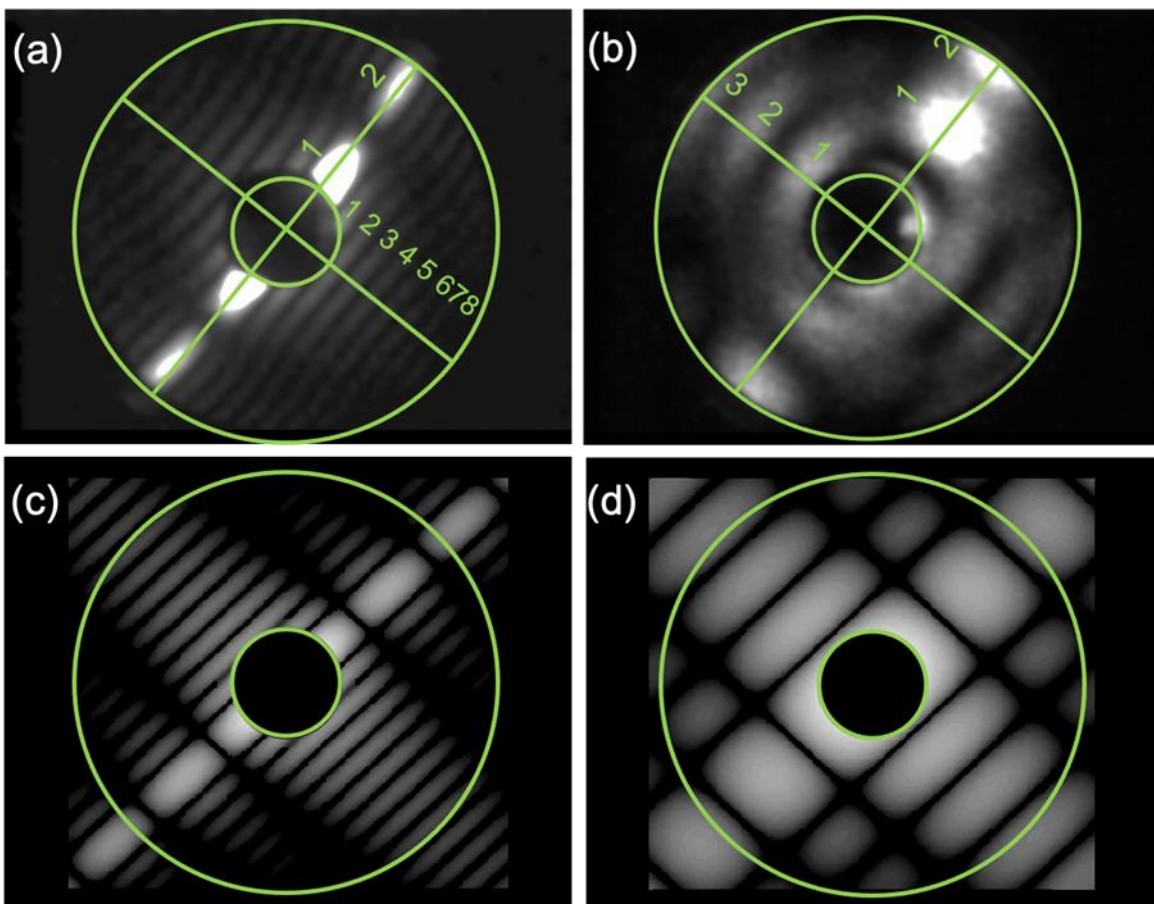

**Figure 2: Examples of 2-D forward scattering patterns of an α-NAD particle (a) and a β-NAD particle (b) captured by SID-3 during homogeneous and heterogeneous NAD nucleation experiments in AIDA, respectively. The two-fold azimuthal symmetry of the patterns indicates particles with two main orthogonal axes like cylinders or spheroids. The ratio of the number of maxima counted along the two symmetry directions in the patterns (diagonal green lines and numbers in (a) and (b)) is used to determine the aspect ratio of the particle. Simulated far field diffraction patterns of 2-D rectangular obstacles with aspect ratios of 4 and 1.5 are shown in (c) and (d), respectively.**




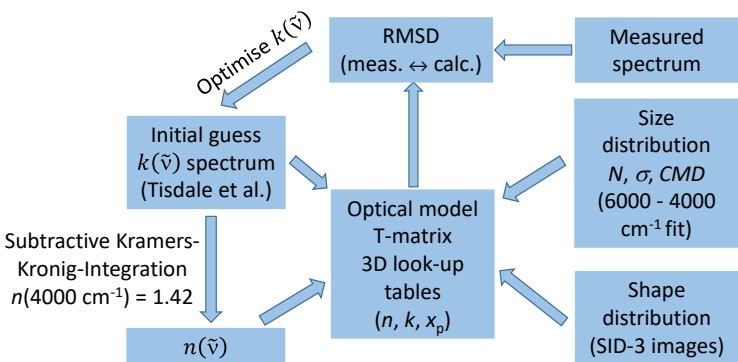


**Figure 3: Retrieval scheme for deriving the optical constants of β-NAD. See Sect. 2.3 for more details and explanation of abbreviations.**



**Figure 4: Comparison of AIDA measurement data for a homogeneous α-NAD nucleation experiment (left part) and a heterogeneous β-NAD nucleation experiment with illite particles (right part). Time zero denotes the start of aerosol injection, and the duration is indicated by the magenta frame. Each part consists of five panels containing the following data sets: (a) AIDA pressure (black) and mean gas temperature (red). (b) Relative humidity with respect to ice (RH$_{ice}$, brown) and saturation ratio with respect to NAD (S$_{NAD}$, green). (c) Size classification of single particles by the welas OPC (blue). (d) Number concentration of crystallised NAD particles (N$_{NAD}$, orange) and number concentration of injected illite particles (N$_{illite}$, pink). (e) Near-backscattering linear depolarisation ratio (δ, grey). See Sect. 3.1 for a detailed description.**



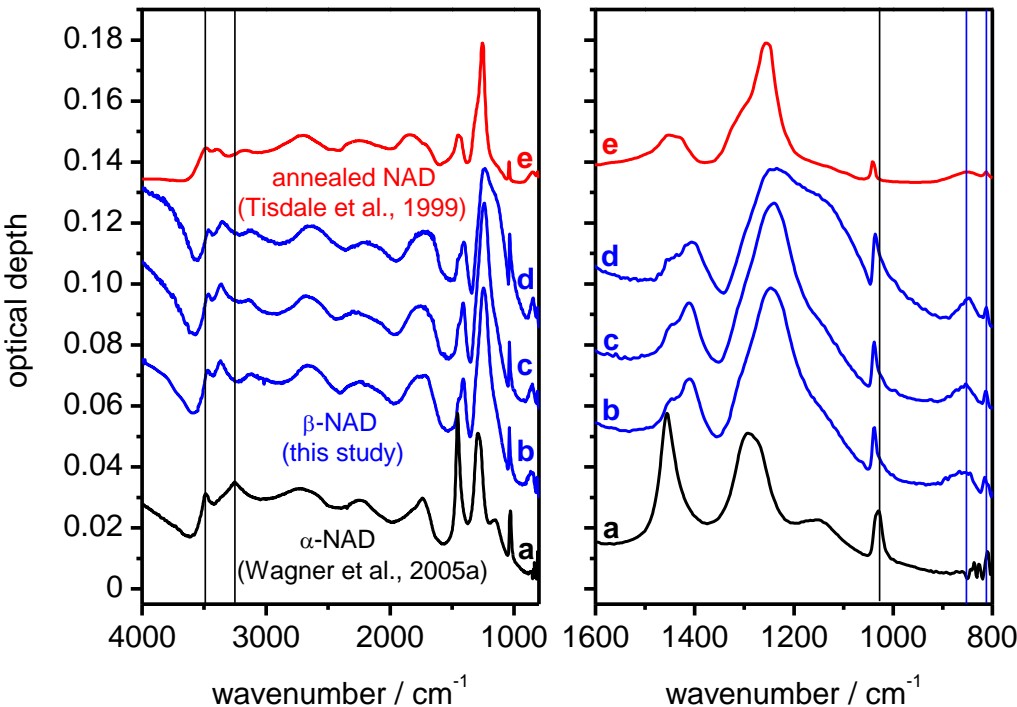

**Figure 5: Comparison of infrared extinction spectra of α- and β-NAD particles. The left part shows the wavenumber range from 4000 to 800 cm⁻¹, while the right part shows an enlarged view from 1600 to 800 cm⁻¹. (a) α-NAD particles generated by shock-freezing a gas mixture of nitric acid and water (Wagner et al., 2005a; Fig. 3 therein, exp. A). (b) – (d) β-NAD particles generated in this study from heterogeneous crystallisation experiments with illite particles at 190 K (b) and 200 K (d), as well as with MgFeSiO₄ particles at 190 K (c). (e) NAD particles formed below 160 K and crystallised between 160 and 221 K during rapid warming (Tisdale et al., 1999; Fig. 4 therein, digitised version of spectrum (E) using WebPlotDigitizer version 4.5). Vertical black and blue lines indicate selected extinction bands characteristic of α- and β-NAD particles, respectively.**



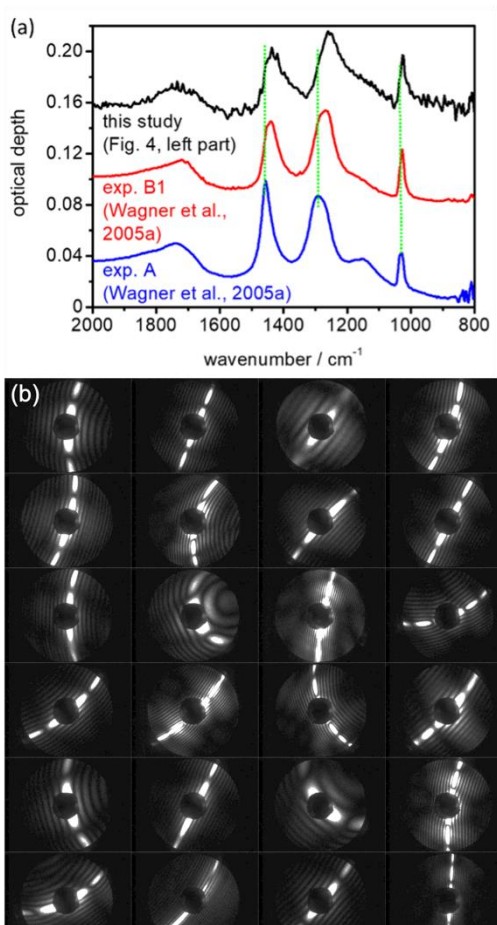

**Figure 6: (a)** Shape-dependent infrared spectral signature of airborne α-NAD crystals. The blue-coloured spectrum was recorded during exp. A from Wagner et al. (2005a) where the α-NAD particles were generated by shock-freezing a gaseous nitric acid – water mixture. In exp. B1 from Wagner et al. (2005a), the α-NAD crystals were formed by homogeneous nucleation from slowly evaporating supercooled $HNO_3/H_2O$ solution droplets at 193 K. A similar long-term crystallisation experiment was performed in this study at 190 K (see Fig. 4, left part), and the infrared spectrum of the nucleated α-NAD crystals is shown in black. Vertical green dotted lines indicate the shifts in band positions between the different spectra. **(b)** SID-3 measurements: Representative two-dimensional scattering patterns of the α-NAD crystals that formed during the long-term crystallisation experiment at 190 K from this study.






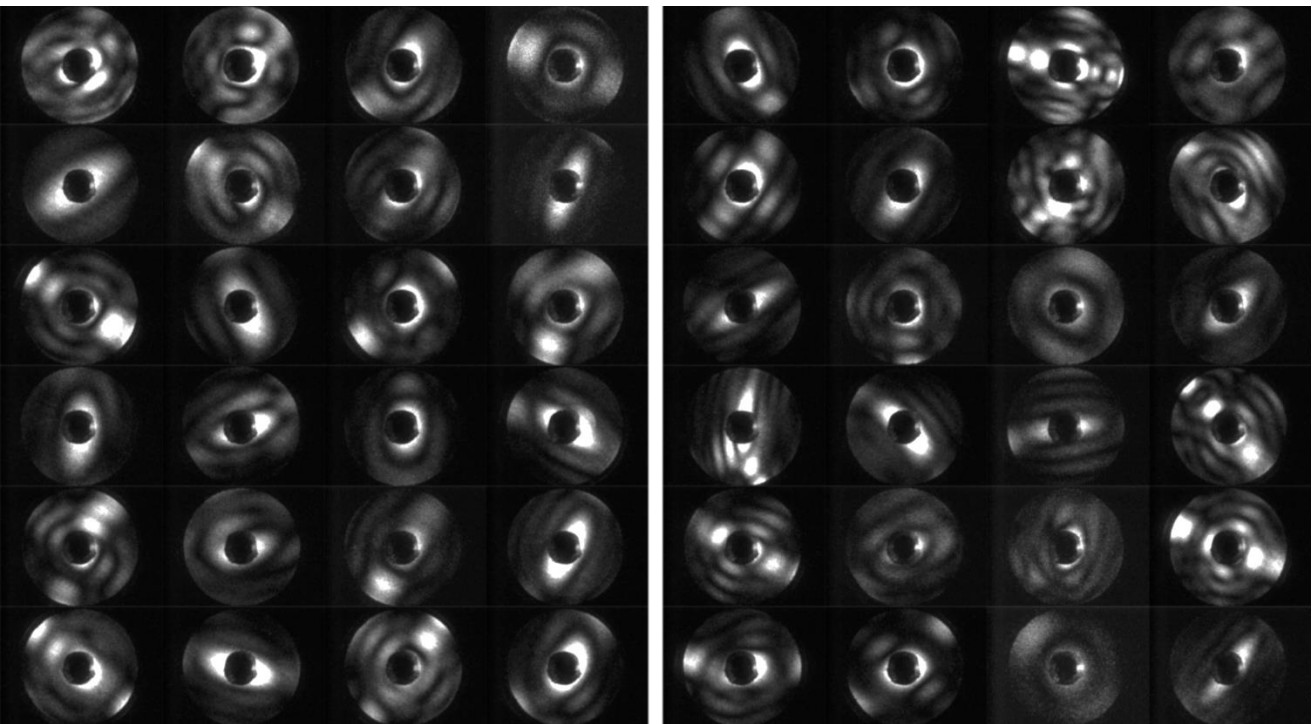

**Figure 7: SID-3 scattering patterns of β-NAD crystals generated during heterogeneous crystallisation experiments with illite (left part) and MgFeSiO₄ (right part) as seed aerosol particles.**


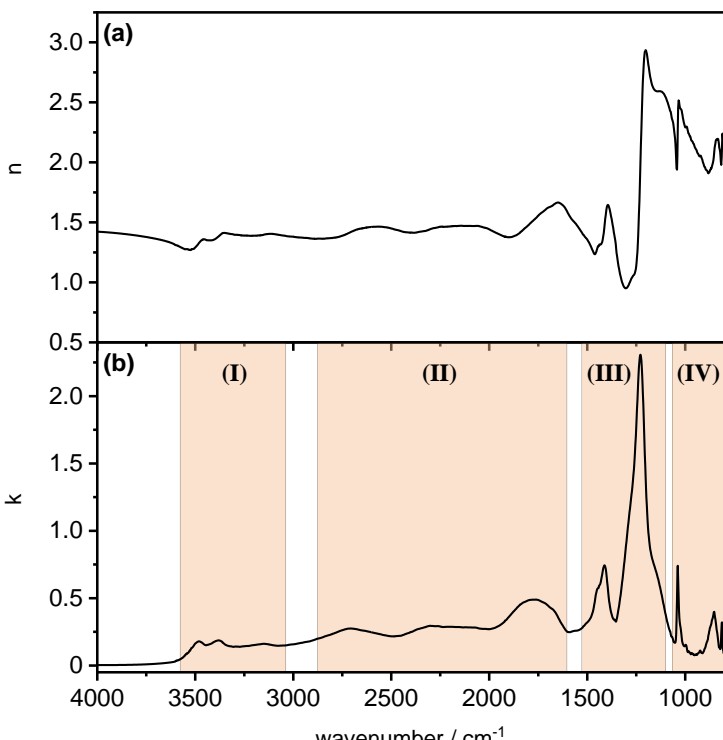

Figure 8: Newly derived data set of mid-infrared optical constants for β-NAD. (a) Real part of the complex refractive index ($n$ spectrum). (b) Imaginary part of the complex refractive index ($k$ spectrum). The four spectral regions highlighted in part (b) are discussed in the text.





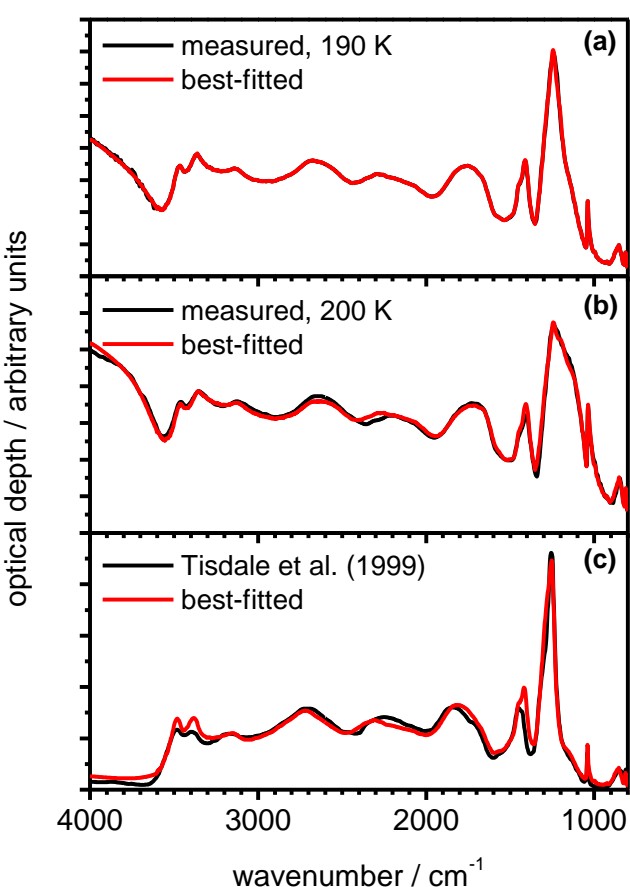

**Figure 9: Measured infrared extinction spectra of β-NAD particles (black) in comparison with best-fitted spectra (red) using the newly derived optical constants displayed in Fig. 8. (a) Heterogeneous crystallisation experiment with illite at 190 K. The measured spectrum was used as template for the iterative adjustment of the optical constants, hence the very close agreement between measurement and calculation. (b) Heterogeneous crystallisation experiment with illite at 200 K. (c) Annealed particles from Tisdale et al. (1999), digitised from trace (E) in Fig. 4 therein. The NAD particles were formed below 160 K and crystallised between 160 and 221 K during rapid warming. For (a) and (b), we used an equivalent mixture of five different aspect ratios in the calculations, whereas the best fit in (c) was obtained for $\phi = 2$.**

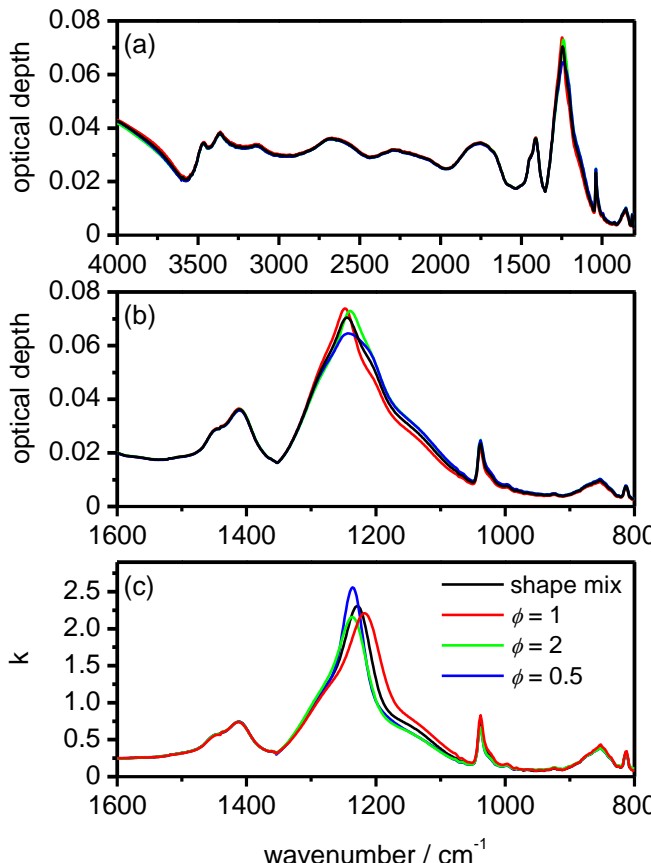


**Figure 10. Shape-dependency of the infrared extinction spectrum and the retrieval results for the *k* spectrum of β-NAD. The colour coding shown in panel (c) applies to all subpanels. (a) Calculation of the β-NAD extinction spectrum from the heterogeneous nucleation experiment with illite at 190 K (see Fig. 9a) for different particle habits. The black spectrum is the same as the best-fitted spectrum in Fig. 9a and was calculated for an equivalent mixture of five aspect ratios, $\phi$ = 0.5, 0.67, 1, 1.5, and 2 (shape mix). For the**
**same underlying size distribution and refractive indices, the additional computations show the spectral habitus when other values are used for the aspect ratio of the particles. (b) Enlarged view of the spectra from panel (a) in the range between 1600 and 800 cm$^{-1}$. (c) Spectral changes in the retrieved *k* spectrum for β-NAD when other values for the aspect ratio of the particles are used instead of the shape mix.**




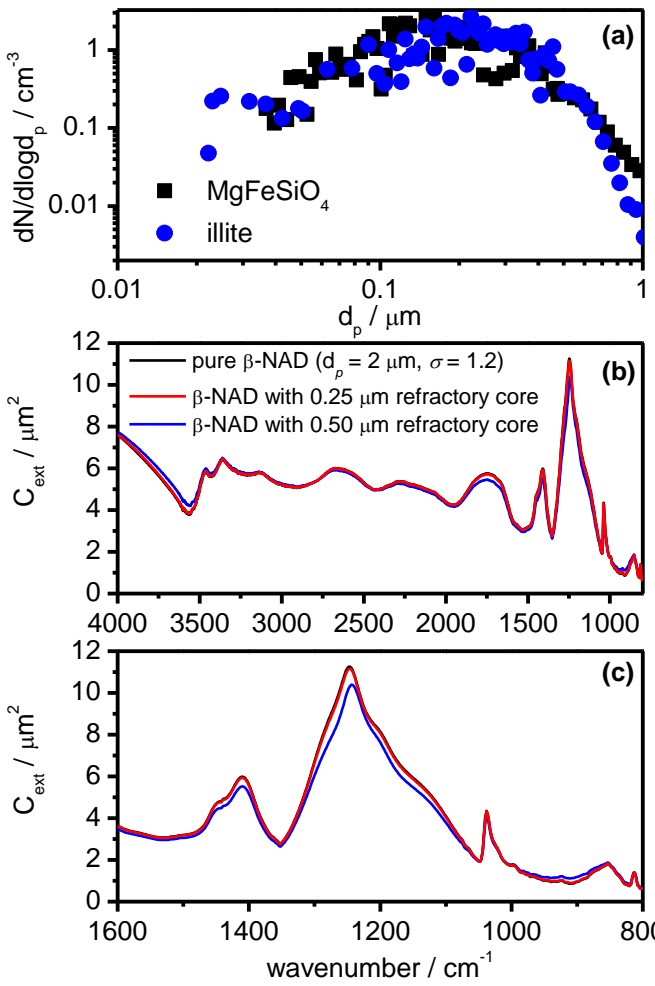

**Figure 11: Influence of refractory inclusions on the infrared extinction spectrum of β-NAD particles. (a) Measurements of the size distribution of pure MgFeSiO₄ and illite particles dispersed with a rotating brush generator and introduced into the AIDA chamber. The size spectra were merged from the recordings of a scanning mobility particle sizer (SMPS, TSI) and an aerodynamic particle spectrometer (APS, TSI). In order to convert measured mobility and aerodynamic diameters into the equal-volume sphere diameter, $d_p$, we used values of 2.6 g cm⁻³ for the particle density and 1.4 for the dynamic shape factor. (b) Infrared extinction cross sections of pure, log-normally distributed spherical β-NAD crystals with a count median diameter of 2 μm and a mode width of 1.2 (black) in comparison with core-shell Mie simulations in which the centre of the β-NAD particles was replaced by spherical refractory inclusions with diameters of 0.25 (red) and 0.50 μm (blue). Note that the black line is almost identical to the red line and therefore difficult to see. (c) Expanded view of the extinction cross sections from panel (b) between 1600 and 800 cm⁻¹.**

