# Peer review of "Particle shapes and infrared extinction spectra of nitric acid dihydrate crystals: Optical constants of the β-NAD modification"

_Atmospheric Chemistry and Physics, 2023_

## Referee Comment (RC1)

**Referee Report on « Particle shapes and infrared extinction spectra of nitric acid dihydrate crystals: Optical constants of the β-NAD modification" authored by R Wagner, A.D. James, V.L. Frankland, O. Möhler, B.J. Murray, J.M. Plane, H. Saathoff, R. Weigel and M. Schnaiter submitted to Atmospheric Physics and Chemistry as manuscript acp-2023-28**

This paper deals with the formation and characterization of a thermodynamically stable polymorph of nitric acid dihydrate using a multidiagnostic approach within the well-known large coolable aerosol chamber (AIDA) equipped with various detection techniques including high-resolution 2D imaging of submicron and supermicron-sized ice particles in the presence of $HNO_3$. This work is convincing, well-done with many quality control benchmarks and informative. The report is well organized, easy to follow, well written and represent a good compromise between sufficient detail and flow of presentation. The new aspect of this work, apart from the surprising discovery of fast growth conditions of β-NAD by heterogeneous nucleation in the presence of proxies of meteoric smoke aerosol particles, resides in the detailed analysis of the scattering properties of resulting solid α- and β-NAD aerosol particles resulting in a marked crystal shape of the two polymorphs. Time will tell whether or not the shapes of elongated needles (α-NAD) or compact spheroids (β-NAD) will explain the atmospheric lifetime or residence time of these aerosol particles in future real observations of given strata within the atmosphere in case β-NAD particles occur at all in nature. I propose the publication of this work in acp once my sparse comments will have triggered a suitable response by the authors.

In what follows I will submit the following comments/questions and remarks concerning the submitted manuscript in the hope to provide a line along which the authors may make changes to the manuscript:

- The displayed FTIR absorption spectra in Figures 5, 6, 8, 9, 10 and 11 are important spectral observables representing key elements in support of α- and β-NAD. However, in comparison to recent spectra, for instance by Iannarelli and Rossi (2015) recorded at nominally identical spectral resolution of 4 cm$^{-1}$ the present spectra show relatively few details in comparison. Have the present FTIR spectra been smoothed in order to suppress small albeit potentially important details? I am aware that recording conditions (T and/or growth conditions, particle size distribution functions, scattering properties, etc.) may lead to minor frequency shifts and small differences in spectral appearance, but what are the reasons for the apparent lack of spectral details in the FTIR spectra? In cases where several FTIR spectra are shown in a stacked manner (for instance in Figure 5 or 6) it is unclear what the displacement of every superimposed spectrum (spectra b to e) is in terms of optical depth or absorption compared to the lowest. I assume that the labelled scale only applies to the lowest displayed absorption spectrum.

- Compared to the referenced precursor studies the present work clearly starts out with liquid droplets of $HNO_3/H_2O$ aerosol, and it is this starting condition that enables the unambiguous observation of β-NAD when meteoric smoke proxies (Illite, $MgFeSiO_4$) are used as seed crystals, otherwise α-NAD is observed for homogeneous freezing without ever ending up as stable β-NAD. This case corresponds to an immersion freezing event triggered by specific (solid) seed aerosol. On the other hand, we and others have exclusively observed the formation of α-NAD in case gas-phase $HNO_3$ is deposited on a macroscopic ice surface without ever observing the conversion from α- to β-NAD. In this case the mechanism might be a case of condensation freezing on PSC II particles that are less prevalent compared to PSC Ia and Ib clouds. Even though the authors state this fact at the end of the article this difference seems to be important enough to alert the reader early on in the report. In

addition, the bifurcation between NAT and NAD depends on the partial pressure (activity) of $HNO_3$ at constant partial pressure of $H_2O$ vapor (Iannarelli and Rossi, 2015): by doubling the flow rate (concentration) of $HNO_3$ we obtain NAD at the expense of $\alpha$-NAT that converts to $\beta$-NAT with increasing temperature. It thus very well may be that NAD (including $\beta$-NAD) may never be accessed depending on the freezing mechanism. Relevant experiments on ice aerosol substrates may be performed in the AIDA chamber in order to test several growth mechanisms, even if the partial pressure of $H_2O$ may be increase somewhat to artificially produce PSC II cloud particles.

- My guess is that both OPC and SID-3 instruments were exposed to low temperatures. Did the authors encounter any temperature problems or other anomalies (calibration) in the vicinity of the measurement ports (inlet or optics)?

- On line 350 what is the definition of the saturation ratio $S_{NAD}$? Ratio of the activities of $HNO_3$ to $H_2O$ water vapor? Please include the expression into the manuscript by mentioning why this was a calculated and not a measured parameter.

- What is the significance of the magenta window in the right and left hand panels of Figure 4: is it to highlight the change in time duration for reaching the peak of rh and the concomitant onset of aerosol particle nucleation?

- The linear depolarization results discussed in lines 491-497 are hard to rationalize in terms of differences of the shapes for $\alpha$- and $\beta$-NAD. What is the physical reason for the three times larger depolarization ratio of $\beta$-NAD vs. $\alpha$-NAD despite the more needle-like shape of the latter? This is somewhat counter-intuitive despite the invoked reference of Zhakarova and Mishchenko (2000) and may have far-reaching consequences for the interpretation of LIDAR atmospheric backscattering signals.

---

## Referee Comment (RC2)

This is a review of the manuscript:

**Particle shapes and infrared extinction spectra of nitric acid dehydrate crystals: Optical constants of the β-NAD modification**

by:

Robert Wagner, Alexander D. James, Victoria L. Frankland, Ottmar Möhler, Benjamin J. Murray, John M. C. Plane, Harald Saathoff, Ralf Weigel, and Martin Schnaiter

in: Atmospheric Chemistry and Physics, 2023.

**Main comments:**

This is an overall well written and structured publication on the measurements of particle shapes, infrared extinction spectra of nitric acid dehydrates (NAD) and optical constants of the metastable form β-NAD in the AIDA aerosol and cloud chamber. The paper is well structured and the main results on the optical constants for β-NAD are especially important for the remote sensing measurements and the characterisation and retrievals of particle size distributions of polar stratospheric clouds (PSCs).

The study is well suited for Atmospheric Chemistry and Physics, but due to its large number of details in laboratory measurements it would even better fit – to my mind - to the sister Journal Atmospheric Measurement Techniques.

The only caveat with the presented study is the open question why the authors have taken so much effort to measure and characterise a PSC particle type in the laboratory so far not measured in the real atmosphere. The authors have presented a discussion addressing this critical question at the end of the manuscript, where they very nicely outline how the 'next' measurements on NAT particles with the AIDA chamber should look like. Hopefully, these kind of measurements will be successful and are coming soon. It might be helpful for the reader to include one summarizing sentence in the abstract and/or introduction section to highlight the importance of NAD lab measurements even though NAT are so far the only measured nitric acid hydrates particles in the atmosphere.

**Minor comments:**

At some place in the manuscript the authors should state that the PSD parameter (e.g. effective radius and or bimodality) are also an important factor for the exact location of the spectral features PSC particles (e.g. Kalicinsky et al. (2021) showed changes in the 820 cm$^{-1}$ NAT feature in direction to a step function for observations under NH polar vortex conditions with radiative transport calculations without taken aspherical particle into account).

L396ff: Here the authors present quite a number of parameters and numbers, maybe a table summarizing the details would be feasible.

L634: Are not Höpfner et al. (2002) and Spang and Remedios (2003) the first publications regarding the NAT identification at 820 cm$^{-1}$ and should be listed here first? Same in the introduction L41, where only Spang and Remedios is mentioned but Höpfner et al. is missing.

**Technical details:**

To my mind the 800-900 cm$^{-1}$ wavenumber range is not very well presented in many figures (6,8,9). It is too tiny, the details are not visible. I have seen this in many publications and is caused by the presentation of a typically huge wave number range. This specific region is of special interest for the remote sensing community due to the characteristic features for nitric acid trihydrate (NAT) at 820 cm$^{-1}$ and the shift of this position and changes in the form of the feature due to various reasons (Woiwode et al. 2016, 2019, Kalicinsky et al. 2021). Maybe the separation in two wavelength region and/or an additional zoom, like in Figure 5 (still very small), for the Fig. 6,8, and 9 would help to show some more details here.

Fig. 6+7: The SID-3 measurements/images are very dark and the diffraction patterns are not very well to see. Can you brighten the images?

**References:**

Höpfner, M., Oelhaf, H., Wetzel, G., Friedl-Vallon, F., Kleinert, A., Lengel, A., Maucher, G., Nordmeyer, H., Glatthor, N., Stiller, G., von Clarmann, T., Fischer, F., Kröger, C., and Deshler, T.: Evidence of scattering of tropospheric radiation by PSCs in mid-IR limb emission spectra: MIPA-B observations and KOPRA simulations, Geophys. Res. Lett., 29, 1278, https://doi.org/10.1029/2001GL014443, 2002.

Kalicinsky, C., Griessbach, S., and Spang, R.: A new method to detect and classify polar stratospheric nitric acid trihydrate clouds derived from radiative transfer simulations and its first application to airborne infrared limb emission observations, Atmos. Meas. Tech., 14, 1893–1915, https://doi.org/10.5194/amt-14-1893-2021, 2021.

Spang, R. and Remedios, J. J.: Observations of a distinctive infrared spectral feature in the atmospheric spectra of polar stratospheric clouds measured by the CRISTA instrument, Geophys. Res. Lett., 30, 1875, https://doi.org/10.1029/2003GL017231, 2003.

Woiwode, W., Höpfner, M., Bi, L., Pitts, M. C., Poole, L. R., Oelhaf, H., Molleker, S., Borrmann, S., Klingebiel, M., Belyaev, G., Ebersoldt, A., Griessbach, S., Grooß, J.-U., Gulde, T., Krämer, M., Maucher, G., Piesch, C., Rolf, C., Sartorius, C., Spang, R., and Orphal, J.: Spectroscopic evidence of large aspherical $\beta$-NAT particles involved in denitrification in the December 2011 Arctic stratosphere, Atmos. Chem. Phys., 16, 9505–9532, https://doi.org/10.5194/acp-16-9505-2016, 2016.

Woiwode, W., Höpfner, M., Bi, L., Khosrawi, F., and Santee, M. L.: Vortex-Wide Detection of Large Aspherical NAT Particles in the Arctic Winter 2011/2012 Stratosphere, Geophys. Res. Lett., 46, 13420–13429, https://doi.org/10.1029/2019GL084145, 2019.

---

## Author Comment (AC1)

**Answer to Michel J. Rossi**

**Thank you for your positive review of our manuscript and the helpful comments. Below we address your individual comments and describe the associated changes (in bold) that have been made in the revised manuscript version.**

This paper deals with the formation and characterization of a thermodynamically stable polymorph of nitric acid dihydrate using a multidiagnostic approach within the well-known large coolable aerosol chamber (AIDA) equipped with various detection techniques including high-resolution 2D imaging of submicron and supermicron-sized ice particles in the presence of HNO3. This work is convincing, well-done with many quality control benchmarks and informative. The report is well organized, easy to follow, well written and represent a good compromise between sufficient detail and flow of presentation. The new aspect of this work, apart from the surprising discovery of fast growth conditions of β-NAD by heterogeneous nucleation in the presence of proxies of meteoric smoke aerosol particles, resides in the detailed analysis of the scattering properties of resulting solid α- and β-NAD aerosol particles resulting in a marked crystal shape of the two polymorphs. Time will tell whether or not the shapes of elongated needles (α-NAD) or compact spheroids (β-NAD) will explain the atmospheric lifetime or residence time of these aerosol particles in future real observations of given strata within the atmosphere in case β-NAD particles occur at all in nature. I propose the publication of this work in acp once my sparse comments will have triggered a suitable response by the authors.

In what follows I will submit the following comments/questions and remarks concerning the submitted manuscript in the hope to provide a line along which the authors may make changes to the manuscript:

• The displayed FTIR absorption spectra in Figures 5, 6, 8, 9, 10 and 11 are important spectral observables representing key elements in support of α- and β-NAD. However, in comparison to recent spectra, for instance by Iannarelli and Rossi (2015) recorded at nominally identical spectral resolution of 4 cm-1 the present spectra show relatively few details in comparison. Have the present FTIR spectra been smoothed in order to suppress small albeit potentially important details? I am aware that recording conditions (T and/or growth conditions, particle size distribution functions, scattering properties, etc.) may lead to minor frequency shifts and small differences in spectral appearance, but what are the reasons for the apparent lack of spectral details in the FTIR spectra? In cases where several FTIR spectra are shown in a stacked manner (for instance in Figure 5 or 6) it is unclear what the displacement of every superimposed spectrum (spectra b to e) is in terms of optical depth or absorption compared to the lowest. I assume that the labelled scale only applies to the lowest displayed absorption spectrum.

As mentioned in line 334, for technical reasons (to improve and speed up the convergence behaviour of the optimisation algorithm) we applied a weak smoothing function when retrieving the *k* spectrum. This was strictly limited to regions with broad absorption signals (e.g. not applied in the regime of the narrow 1038 and 812 $cm^{-1}$ modes) and affected the *k* spectrum obtained (Fig. 8) and the calculations performed with this data set (Figs. 9 - 11). What we actually missed to report is that we corrected all the displayed measured spectra for imbalances in the gas phase $CO_2$ concentration between the reference runs before aerosol injection and the later sample runs in the wavenumber range from 2400 to 2280 $cm^{-1}$. We have added this information in line 202:

"All measured spectra displayed in this article were corrected for imbalances in the gas phase $CO_2$ concentration between the reference runs before aerosol injection and the later sample runs in the wavenumber range from 2400 to 2280 $cm^{-1}$."

A correction for water vapour signals (at 1600 and 3600 $cm^{-1}$) was not applied. The displayed NAD spectra were recorded at times when the transient increase in relative humidity after aerosol addition (see Fig. 4) had already subsided, so that the spectra were measured at similar relative humidity compared to the reference runs. This and the very low absolute water vapour concentration at low temperatures in the AIDA chamber made the $H_2O$ correction obsolete.

The observation of higher feature detail in the spectra of Iannarelli and Rossi (2015) could also be due to the fact that these spectra show a progressive change in the signatures as $HNO_3$ was deposited on the ice films, and that the spectra also represent a superposition of two species, ice and NAT/NAD. Deconvolution was used to derive the pure NAD/NAT spectra, which could give rise to some features, as the pure substances in a mixture can undergo small frequency shifts.

We indeed missed an accurate description of the displacement applied to the stacked spectra in Figs. 5 and 6, thank you for pointing this out. Yes, the scale only refers to the lowermost spectrum and is therefore misleading for the other spectra. We therefore propose to change the labelling similar to the representation chosen by Grothe et al. (2004): We remove the label numbers and indicate by a bar in the panel the height of e.g. 0.02 optical depth (OD) units. The revised Fig. 5 would then look like (the upper part of Fig. 6 will be changed accordingly):

[Figure]

**Added statement at the end of the figure caption:**

**"Individual spectra have been offset for clarity. The digitised absorbance spectrum (e) does not adhere to the optical depth scale and has been arbitrarily scaled to allow comparison with the AIDA measurements."**

• Compared to the referenced precursor studies the present work clearly starts out with liquid droplets of HNO3/H2O aerosol, and it is this starting condition that enables the unambiguous observation of β-NAD when meteoric smoke proxies (Illite, MgFeSiO4) are used as seed crystals, otherwise α-NAD is observed for homogeneous freezing without ever ending up as stable β-NAD. This case corresponds to an immersion freezing event triggered by specific (solid) seed aerosol. On the other hand, we and others have exclusively observed the formation of α-NAD in case gas-phase HNO3 is deposited on a macroscopic ice surface without ever observing the conversion from α- to β-NAD. In this case the mechanism might be a case of condensation freezing on PSC II particles that are less prevalent compared to PSC Ia and Ib clouds. Even though the authors state this fact at the end of the article this difference seems to be important enough to alert the reader early on in the report. In addition, the bifurcation between NAT and NAD depends on the partial pressure (activity) of HNO3 at constant partial pressure of H2O vapor (Iannarelli and Rossi, 2015): by doubling the flow rate (concentration) of HNO3 we obtain NAD at the expense of α-NAT that converts to β-NAT with increasing temperature. It thus very well may be that NAD (including β-NAD) may never be accessed depending on the freezing mechanism. Relevant experiments on ice aerosol substrates may be performed in the AIDA chamber in order to test several growth mechanisms,

even if the partial pressure of H2O may be increase somewhat to artificially produce PSC II cloud particles.

**We are happy to take up this suggestion, i.e. to emphasise the different nucleation pathways already at the beginning of the article, and have added the following paragraph at the end of our introduction in line 141:**

**"In anticipation of a later more detailed discussion of possible NAD/NAT nucleation mechanisms, we would like to emphasise here that our experiments address a particular heterogeneous nucleation process for β-NAT, i.e. immersion freezing induced by specific solid seed aerosol particles. This differs from other studies such as Iannarelli and Rossi (2015), where the deposition of $HNO_3$ vapour on pure $H_2O$ ice led to the almost barrier-free growth of α-NAT and NAD. Iannarelli and Rossi (2015) also showed that the type of particles formed was sensitive to the partial pressure of $HNO_3$, with lower values favouring the formation of α-NAT. Depending on the freezing mechanism, neither α- nor β-NAD could therefore be accessible at all."**

• My guess is that both OPC and SID-3 instruments were exposed to low temperatures. Did the authors encounter any temperature problems or other anomalies (calibration) in the vicinity of the measurement ports (inlet or optics)?

**SID-3 is an airborne instrument that has been used frequently in aircraft campaigns in the stratosphere and is therefore robust for use at low temperatures (e.g. some optical components are heated to prevent condensation). No special adjustments were required for its use in the AIDA chamber. As the only precaution, when the SID-3 electronics were switched off and no longer generated internal heat, but the instrument was still exposed to low temperatures, we heated the stainless steel container in which the instrument was installed. We will include the information that the container was heatable in the revised manuscript version.**

**The OPC works with fibre optic technology, i.e. only the measurement cell (sensor unit) is exposed to low temperatures, while the electronics with light source and photomultiplier detector are located outside the insulating housing of the AIDA chamber (connected with optical fibres). Its use is therefore also specified for very low temperatures. We added a respective statement in line 228:**

**"The electronic unit of the OPC with light source and photomultiplier detector is located outside the insulating housing and is connected to the sensor unit with optical fibres."**

• On line 350 what is the definition of the saturation ratio SNAD? Ratio of the activities of HNO3 to H2O water vapor? Please include the expression into the manuscript by mentioning why this was a calculated and not a measured parameter.

We propose to extend our description in line 350 as follows by now first explaining the definition of $S_{NAD}$ and then our calculation method in more detail.

"$S_{NAD}$ is the saturation ratio of the liquid phase with respect to solid NAD. Explicitly, it is the quotient between the activity product of the ions ($H^+$, $NO_3^-$) and the solvent ($H_2O$) in the liquid phase, i.e. $a(H^+) \cdot a(NO_3^-) \cdot a^2(H_2O)$, and the respective activity product in a solution saturated with respect to solid NAD ($K_s$) (Salcedo et al., 2001). Assuming that the liquid and the gas phase are at equilibrium, we first calculated the above activity product for the temperature and relative humidity conditions prevalent in the AIDA chamber with the E-AIM model (Clegg et al., 1998; Massucci et al., 1999). The formation of solid phases was hereby prevented. Afterwards, the activity product for NAD-saturated conditions, $K_s$, was computed according to the temperature-dependent function given in Eq. (A9) of Massucci et al. (1999). The ratio of these two activity products then yielded $S_{NAD}$."

More directly, the E-AIM model could also be fed with the composition of the nitric acid solution droplets obtained from the FTIR spectra to compute the activity product in the liquid phase (Stetzer et al., 2006; Möhler et al., 2006a). However, due to the rapid growth of β-NAD in the heterogeneous nucleation experiments, the recorded infrared spectra quickly showed an overlap of the signatures of the liquid $HNO_3$/$H_2O$ solution droplets and the solid β-NAD crystals. An accurate determination of droplet composition was therefore not possible, so we constrained the E-AIM model to the measured relative humidity instead.

• What is the significance of the magenta window in the right and left hand panels of Figure 4: is it to highlight the change in time duration for reaching the peak of rh and the concomitant onset of aerosol particle nucleation?

The magenta-coloured frames indicate the duration of the injection of the mixed $HNO_3$ and $H_2O$ carrier gases (in the case of the β-NAD experiment together with the seed aerosol particles). This is mentioned in line 348 in the article and in the figure caption.

• The linear depolarization results discussed in lines 491-497 are hard to rationalize in terms of differences of the shapes for α- and β-NAD. What is the physical reason for the three times larger depolarization ratio of β-NAD vs. α-NAD despite the more needle-like shape of the latter? This is somewhat counter-intuitive despite the invoked reference of Zhakarova and Mishchenko (2000) and may have far-reaching consequences for the interpretation of LIDAR atmospheric backscattering signals.

The scattering properties of needlelike and platelike particles are indeed "unique" and, therefore, "the weak depolarization capability of highly aspherical particles should be

taken into account during analysis of lidar depolarization measurements" (two quotes from the article by Zakharova and Mishchenko (2000). It is difficult to put this behaviour into a simple physical picture. Zakharova and Mishchenko (2000) found that some scattering parameters for the needlelike and platelike particles, such as the phase functions and the asymmetry parameters, are similar to those of surface-equivalent spheres. In other words, these parameters are determined by the value of the size parameter of the sphere having the same projected area as the aspherical particle. Other scattering parameters, however, such as the extinction efficiencies and the depolarisation ratios (linear and circular), are more sensitive to the value of the size parameter along the smallest particle axes, which can assume values below unity in the case of strong deviations from the spherical shape, even if the surface-equivalent-sphere size parameters are much larger. For these quantities, the scattering behaviour of needlelike and platelike particles is then typical of Rayleigh scatterers, i.e. they show much lower extinction efficiencies and lower depolarisation ratios compared to compact particles with the same surface-equivalent-sphere size parameters. This then leads to the important conclusion that the magnitude of the depolarisation ratio cannot be simply related to the degree of asphericity. We will expand the reference to Zakharova and Mishchenko (2000) (line 492) in the revised manuscript text as follows:

"Using ice crystals as an example, Zakharova and Mishchenko (2000) have shown that wavelength-sized needle- and plate-like particles, modelled as spheroids with aspect ratios of 0.05 and 20, have unique scattering properties. While some scattering parameters, such as the phase functions and the asymmetry parameters, are similar to those of surface-equivalent spheres, other quantities, such as the extinction efficiencies and the depolarisation ratios (linear and circular), resemble those of Rayleigh particles. In particular, such needle- and plate-like particles cause much less backscattering linear depolarisation than surface-equivalent particles with moderate aspect ratios of 0.5 and 2."

---

## Author Comment (AC2)

**Answer to Anonymous Referee #2**

**Thank you for your positive evaluation of our article and the helpful comments. Below we address your individual comments and describe the associated changes (in bold) that have been made in the revised manuscript version.**

Main comments:

This is an overall well written and structured publication on the measurements of particle shapes, infrared extinction spectra of nitric acid dehydrates (NAD) and optical constants of the metastable form β-NAD in the AIDA aerosol and cloud chamber. The paper is well structured and the main results on the optical constants for β-NAD are especially important for the remote sensing measurements and the characterisation and retrievals of particle size distributions of polar stratospheric clouds (PSCs).

The study is well suited for Atmospheric Chemistry and Physics, but due to its large number of details in laboratory measurements it would even better fit – to my mind - to the sister Journal Atmospheric Measurement Techniques.

**We can understand this train of thought, but would still like to promote a publication in Atmospheric Chemistry and Physics (ACP). One of our earlier articles on the optical constants of ammonium nitrate was indeed transferred to Atmospheric Measurement Techniques (AMT) (https://amt.copernicus.org/articles/14/1977/2021/). This decision was understandable because that article focused on the procedure for determining the optical constants, while the underlying experiments on the formation of ammonium nitrate had already been described in another article. In our opinion, this is clearly different in the present work. Here we do not only describe the retrieval procedure, but present a set of novel laboratory experiments on the immersion freezing of HNO$_3$/H$_2$O solution droplets that led to a surprising result, namely the rapid growth of the β-NAD phase. We also provide a detailed literature review of previous studies on NAD/NAT nucleation and an outlook on future experiments planned in our laboratory. From our point of view, the article therefore goes beyond the scope of AMT, which deals with the development, comparison and validation of measurement instruments, and ACP is a much better fit.**

The only caveat with the presented study is the open question why the authors have taken so much effort to measure and characterise a PSC particle type in the laboratory so far not measured in the real atmosphere. The authors have presented a discussion addressing this critical question at the end of the manuscript, where they very nicely outline how the 'next' measurements on NAT particles with the AIDA chamber should look like. Hopefully, these kind of measurements will be successful and are coming soon. It might be helpful for the reader to include one summarizing sentence in the abstract and/or introduction section to highlight the

importance of NAD lab measurements even though NAT are so far the only measured nitric acid hydrates particles in the atmosphere.

**This is a good suggestion. In the introduction we addressed the importance of metastable states with reference to Ostwald's step rule and the potentially lower nucleation barrier of NAD compared to NAT, but we failed to do so in the abstract. We propose to add a summarising statement at the end of the abstract as follows:**

**"While direct evidence for the existence of metastable NAD in the polar stratosphere is still lacking, our experiments add to the wealth of previous laboratory studies that have identified various conditions for the rapid growth of metastable compositions. In the atmosphere, these could be intermediate states that transform into thermodynamically stable NAT on longer time scales in aged PSCs."**

Minor comments:

At some place in the manuscript the authors should state that the PSD parameter (e.g. effective radius and or bimodality) are also an important factor for the exact location of the spectral features PSC particles (e.g. Kalicinsky et al. (2021) showed changes in the 820 cm-1 NAT feature in direction to a step function for observations under NH polar vortex conditions with radiative transport calculations without taken aspherical particle into account).

**Thank you for pointing out this study, which of course deserves to be mentioned in the introduction. We added the information in line 91:**

**"Note that the particle size distribution parameters of $\beta$-NAT can also change the spectral features in the range from 810 to 820 $cm^{-1}$, even if the simulations are limited to spherical particles (Kalicinsky et al., 2021). Increasing the median diameter transforms the 820 $cm^{-1}$ peak into a shifted peak and further into a step-like signature (Kalicinsky et al., 2021). But only highly aspherical particle shapes could also explain the unexpectedly large optical diameters (in some cases larger than 20 $\mu$m) of $HNO_3$-containing particles detected with aircraft-borne optical in situ instruments (forward scattering and optical array imaging probes) (Molleker et al., 2014)."**

L396ff: Here the authors present quite a number of parameters and numbers, maybe a table summarizing the details would be feasible.

**Yes, we have inserted a reference to a new table in line 367 in which we have summarised the start parameters of the homogeneous and the heterogeneous nucleation experiment shown in Fig. 4:**

**"Relevant start parameters of the two nucleation experiments are summarised in Table 1."**

**Table 1: Start parameters of the homogeneous α-NAD and the heterogeneous β-NAD nucleation experiment shown in Fig. 4. AIDA temperature (*T*), pressure (*p*), and number concentration of added illite particles ($N_{illite}$) were directly measured. Mass concentration ($m_{NA}$) and composition ($wt\%_{NA}$) of the $HNO_3/H_2O$ solution droplets were retrieved from the FTIR spectra. The log-normal parameters of the size distribution of the $HNO_3/H_2O$ solution droplets ($N_{NA}$, $\sigma_{NA}$, $CMD_{NA}$) were constrained using the welas OPC measurements.**

| Exp. type | $T$ (K) | $p$ (hPa) | $N_{NA}$ (cm$^{-3}$) | $\sigma_{NA}$ | $CMD_{NA}$ (μm) | $m_{NA}$ (mg/m$^3$) | $wt\%_{NA}$ | $N_{illite}$ (cm$^{-3}$) |
|---|---|---|---|---|---|---|---|---|
| Hom (α-NAD) | 190.7 | 994 | 25000 | 1.2 | 0.5 | 2.7 | 50 | – |
| Het (β-NAD) | 190.3 | 1012 | 20000 | 1.2 | 0.5 | 2.2 | 53 | 100 |

L634: Are not Höpfner et al. (2002) and Spang and Remedios (2003) the first publications regarding the NAT identification at 820 cm$^{-1}$ and should be listed here first? Same in the introduction L41, where only Spang and Remedios is mentioned but Höpfner et al. is missing.

**Thanks for pointing this out. We will add Höpfner et al. (2002) to line 41 and both Höpfner et al. (2002) and Spang and Remedios (2003) to line 634.**

Technical details: To my mind the 800-900 cm$^{-1}$ wavenumber range is not very well presented in many figures (6,8,9). It is too tiny, the details are not visible. I have seen this in many publications and is caused by the presentation of a typically huge wave number range. This specific region is of special interest for the remote sensing community due to the characteristic features for nitric acid trihydrate (NAT) at 820 cm$^{-1}$ and the shift of this position and changes in the form of the feature due to various reasons (Woiwode et al. 2016, 2019, Kalicinsky et al. 2021). Maybe the separation in two wavelength region and/or an additional zoom, like in Figure 5 (still very small), for the Fig. 6,8, and 9 would help to show some more details here.

**This is a good suggestion. With the spectra for α-NAD shown in Fig. 6, we deliberately wanted to focus on the range of the nitrate doublet between 1500 and 1100 cm$^{-1}$, where the shape-dependent variations are most obvious. However, for the new spectral measurements and retrievals of β-NAD, we agree that it is extremely useful to better represent the wavenumber range between 900 and 800 cm$^{-1}$. Since the best-fitted spectra shown in Fig. 9 make direct use of the optical constants shown in Fig. 8, it seems redundant to add a zoom to both figures. Therefore, we have concentrated on Fig. 9 and added separate panels for the range from 900 to 800 cm$^{-1}$ for all three case studies.**

**New Fig. 9:**

[Figure]

Fig. 6+7: The SID-3 measurements/images are very dark and the diffraction patterns are not very well to see. Can you brighten the images?

**Yes, we will try to brighten the images a bit and also pay special attention to the quality of these figures during the production process.**